# Breathing dissipative solitons in optical microresonators

E. Lucas[1], M. Karpov[1], H. Guo[1], M.L. Gorodetsky[2,3] & T.J. Kippenberg[1]

Dissipative solitons are self-localised structures resulting from the double balance of dispersion by nonlinearity and dissipation by a driving force arising in numerous systems. In Kerr-nonlinear optical resonators, temporal solitons permit the formation of light pulses in the cavity and the generation of coherent optical frequency combs. Apart from shape-invariant stationary solitons, these systems can support breathing dissipative solitons exhibiting a periodic oscillatory behaviour. Here, we generate and study single and multiple breathing solitons in coherently driven microresonators. We present a deterministic route to induce soliton breathing, allowing a detailed exploration of the breathing dynamics in two microresonator platforms. We measure the relation between the breathing frequency and two control parameters—pump laser power and effective-detuning—and observe transitions to higher periodicity, irregular oscillations and switching, in agreement with numerical predictions. Using a fast detection, we directly observe the spatiotemporal dynamics of individual solitons, which provides evidence of breather synchronisation.

[1] IPHYS, École Polytechnique Fédérale de Lausanne (EPFL), CH-1015 Lausanne, Switzerland. [2] Russian Quantum Centre, Skolkovo 143025, Russia. [3] Faculty of Physics, M.V. Lomonosov Moscow State University, 119991 Moscow, Russia. E. Lucas and M. Karpov contributed equally to this work.  Correspondence and requests for materials should be addressed to T.J.K. (email: tobias.kippenberg@epfl.ch)

issipative solitons are localised structures, occurring in a wide variety of dissipative nonlinear systems in plasma physics, matter waves, optics, chemistry and biology[1]. Recently, the generation of temporal dissipative Kerr solitons (DKS) through parametric conversion in optical micro-resonators[2] triggered a substantial interest. Indeed, DKS constitute a way to generate coherent optical frequency combs with large repetition rates in the range of tens to hundreds of gigahertz, having a vast application potential. They have already been employed in a growing number of proof-of-concept experiments, including coherent terabit telecommunications[3], dual-comb spectroscopy[4] and for the realisation of a microwave-to-optical link via self-referencing[5, 6]. In parallel, nonlinear microresonators appeared as a suitable platform to study the properties and dynamics of nonlinear systems. Despite their apparent simplicity, microresonators possess a rich variety of stable inhomogeneous solutions. As reported both in numerical simulations[7–10] and in experiments[2, 11–13], depending on its dispersion parameters, a resonator can sustain bright dissipative Kerr solitons[2, 12, 13], Turing patterns[14], soliton crystals[15] or dark pulses (platicons)[11, 16]. Some of these localised patterns can exhibit a rich panel of dynamical instabilities. In particular, bright DKS, supported in the case of anomalous group velocity dispersion (GVD), can undergo breathing, i.e., a periodic variation in their duration and amplitude[17–20]. Similar to Kuznetsov-Ma[21, 22] and Akhmediev breathers[23] in conservative nonlinear systems, breathing dissipative solitons are related to the Fermi-Pasta-Ulam recurrence[19]—a paradoxical evolution of nonlinearly coupled oscillators, that periodically return to the initial state[24]. Characterising the breather regime is not only of fundamental interest, but important for applications. The accurate knowledge of the conditions for breathing allows the prevention of soliton instabilities and extreme events such as solitons collisions[25], ensuring a stable operation of DKS-based microresonator devices, and avoiding excess noise[26].

Breathing dissipative solitons were first demonstrated in fibre cavities by Leo et al.[18]. However, the experimental observation of breathers in optical microresonators has posed a significant challenge due to the non-trivial soliton generation process[2, 13], the thermal nonlinearity that may impact the effective laser detuning[27, 28] and high repetition rates (>10 GHz) that make direct time-resolved observations difficult. Nevertheless, micro-resonators remain an attractive platform to study breathers, as their very high finesse allows easy access to strong driving regimes where complex instabilities are predicted to occur[18]. Furthermore, the fact that the timescale of the instabilities scales with the relatively long photon lifetime, combined with the ability to generate a single soliton in the cavity suggests that the soliton dynamics can still be captured with a sampling rate much smaller than the repetition rate. Recently, the observation of breathers in microresonators was reported, evidencing the link with the Fermi-Pasta-Ulam recurrence[19] in particular, as well as a first attempt at characterising breather dynamics, in which the oscillation frequency (breathing frequency) was shown to depend on the pump detuning, but the direct relation was not unambiguously determined.[20]

Here, we present a comprehensive analysis of breathing dissipative solitons in microresonators. First, we demonstrate a deterministic route to access and characterise breathing solitons in two microresonator platforms: crystalline $MgF_2$ whispering gallery mode resonators and $Si_3N_4$ integrated microresonators. Second, owing to a newly introduced probing method[28], we directly measure the operating conditions, allowing a detailed exploration of the breathing regime, revealing the relation between breathing frequency and the driving laser parameters. Third, we map the breathers' existence range and study its

dependence on the pump power. Fourth, we present time-resolved observations of the intracavity pattern evolution in an optical microresonator. This enables the behaviour of individual soliton pulses to be tracked, even with several solitons in the cavity, revealing non-stationary breathing dynamics as well as evidence of breather synchronisation.

## Results

**Deterministic access to dissipative breathing solitons.** The nonlinear dynamics of the optical field in continuous wave (CW)-laser-driven microresonators in the presence of the Kerr non-linearity can be very accurately described using a system of nonlinear coupled mode equations[29, 30], demonstrating almost perfect correspondence with experimental data[26]. This system of equations may be considered as a discrete Fourier transform of the damped driven nonlinear Schrödinger equation (NLSE)[2]:

$$i\frac{\partial\Psi}{\partial\tau} + \frac{1}{2}\frac{\partial^2\Psi}{\partial\theta^2} + |\Psi|^2\Psi = (-i + \zeta_0)\Psi + if. \tag{1}$$

Here $\Psi(\tau, \theta)$ is the normalised intracavity waveform, $\theta$ is the dimensionless longitudinal coordinate, and $\tau$ the normalised time. Equation (1) is usually termed in optics as the Lugiato-Lefever equation (LLE)[31], where a transverse coordinate is used instead of a longitudinal one in our case. The nonlinear dynamics of the system is determined by two parameters: the normalised pump amplitude $f$ and detuning $\zeta_0$, defined as[2]:

$$f = \sqrt{\frac{8g\eta P_{in}}{\kappa^2\hbar\omega_0}}, \quad \zeta_0 = \frac{2\delta\omega}{\kappa}, \tag{2}$$

where $\kappa$ denotes the loaded resonator linewidth ($Q = \omega_0/\kappa$, loaded quality factor), $\eta = \kappa_{ex}/\kappa$ the coupling coefficient, $P_{in}$ the pump power, $\omega_0$ the pumped resonance frequency and $\delta\omega = 2\pi\delta = \omega_0 - \omega_p$ is the detuning of the pump laser from this resonance, counted positive for a red-detuned laser. The non-linearity is described via $g = \hbar\omega_0^2 cn_2/n_0^2 V_{eff}$ giving the Kerr frequency shift per photon, with the effective refractive index $n_0$, nonlinear refractive index $n_2$, and the effective optical mode volume $V_{eff}$.

A similar equation was first analysed in plasma physics[32, 33]. These early studies demonstrated that stable soliton attractors exist within a certain range of effective detuning in the red-detuned regime (green area in Fig. 1a). In addition, it was also shown that, oscillating time-periodic solitons (i.e., breathers) and chaotic states are possible[7]. Extensive numerical analysis and charting of the parameter space of Eq. (1)[7–9, 18] revealed that the breathing region is located close to the low-detuning boundary of the soliton existence range (red area in Fig. 1a). Theoretically the transition from stationary to oscillating DKS results from a Hopf bifurcation that arises above a certain pump power level[18]. The simulated temporal evolution of the amplitude profile of a breathing soliton is displayed in Fig. 1b, showing that the DKS compresses and stretches periodically. The spectral envelope also reflects this effect (Fig. 1c). Breathers are also known to radiate weakly decaying dispersive waves[34] that induce sidebands on the optical spectrum.

We apply the so-called laser backward tuning method[28] in order to deterministically access the breathing regime of a single soliton in a microresonator. A similar approach was recently employed independently in fibre cavities[35]. In this approach, a stationary multiple-soliton state is first generated by sweeping the continuous wave (CW) driving laser forward (towards longer wavelengths) over the pumped resonance and stopping on the effectively red-detuned side, where solitons are sustained[2]. Second, the driving laser is tuned backward (towards shorter

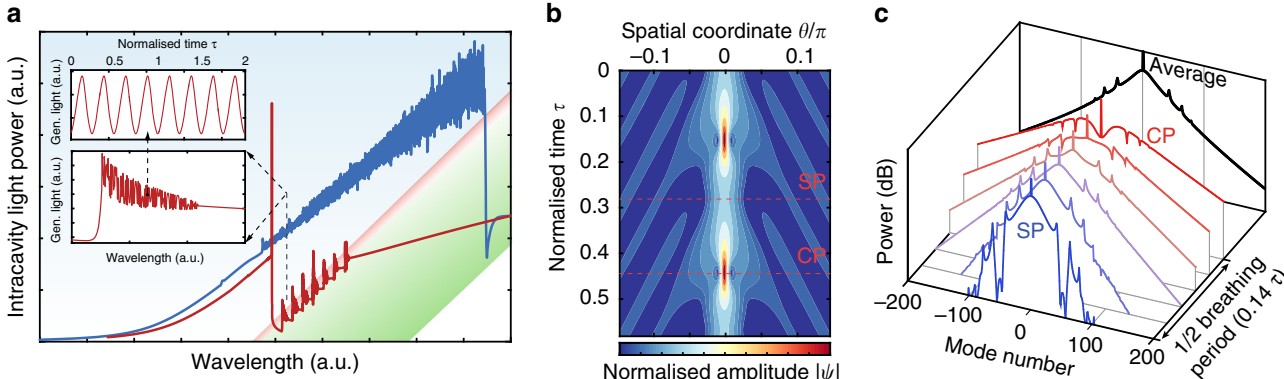

**Fig. 1** Numerical simulations. **a** Simulation of the intracavity power, illustrating the backward tuning method used to trigger breathing. The generation of a stable multiple-soliton state is achieved by forward tuning of the pump laser (*blue curve*). The backward tuning is applied next (*red curve*) in order to reach the low-detuning boundary of the soliton existence range where the breathing regime (increased noise) and switching effect (step features) occur, allowing to transition to the single-soliton state. The *blue shading* corresponds to the region where modulation instability occurs, the green marks stationary solitons existence and the *red area* indicates breathing. The *lower* inset details the single soliton breathing and switching during the backward tuning. The *upper* inset shows the oscillations of the power for a fixed laser frequency in the single-soliton state ($f = 21.5$, $\zeta_0 = 27.3$). **b** Temporal evolution of a breathing dissipative soliton over one breathing period illustrating the periodic compression (CP) and stretching (SP), both indicated by a *red dashed line* and the emission of waves in the background. **c** Corresponding optical spectrum of the intracavity pattern over half a period, showing the evolution between the maximal stretching and compression instants (SP and CP). On average, over one period, the spectrum features a triangular shape (in log scale). The emitted waves are causing the sidebands in the spectrum

wavelength), thus reducing the effective detuning. Thanks to the microresonator thermal nonlinearity that lifts the fundamental degeneracy of multiple-soliton states, this approach was shown to enable the reduction of the intracavity soliton number, via successive switchings to states with a smaller soliton number, thus enabling single soliton access[28]. Figure 1a shows a simulation of this excitation scheme in the $Si_3N_4$ microresonator, including the thermal effects. Forward and backward tuning stages are indicated with blue and red lines correspondingly. The system experiences a series of consecutive switchings, as reflected by the stair-like trace of the intracavity power. We observed that the breathing regime is characterised by oscillations in the intracavity power that occur in the vicinity of the switching points in each step (inset in Fig. 1a). Importantly, the breathing dynamics can be unambiguously characterised only in the single-soliton state, where interactions among different solitons (in a multiple-soliton state) are avoided.

**Experimental identification of breathing**. We experimentally verified our approach in both platforms, in which a breathing single soliton was generated using the backward tuning method (see experimental setup Fig. 2a). Despite significant differences in the resonators properties (Q factor, free spectral range (FSR), material dispersion and nonlinearity, see Methods section), both systems behave qualitatively similarly when approaching and entering the breathing regime. Figure 2c, d show the experimental evolution of the generated light power of a single soliton in $Si_3N_4$ and $MgF_2$ optical resonators when the backward tuning is applied. This signal is obtained by detecting the out-coupled light, after attenuation of the strong pump laser with a narrow fibre Bragg grating notch filter. In both cases the system evolved from a stationary DKS on the right of each trace, to breathing DKS, and finally switched to a homogeneous background, without soliton. In both platforms, reaching the breathing regime coincides with a progressively increased amplitude noise of the generated light power. A detailed measurement (with an increased sampling rate) reveals that the power is oscillating, as shown in the inset of Fig. 2d.

The oscillatory nature of the out-coupled pulse train in the breathing state can also be characterised by measuring its radio frequency (RF) spectrum on an electronic spectrum analyser

(ESA). Figure 2e, g shows the RF spectra of the generated light for stationary and breathing DKS in both optical resonator platforms, at the points marked in Fig. 2c, d. The stationary soliton state (blue traces) corresponds to a low-noise state of the system, while the breathing state exhibits sharp tones indicating the fundamental breathing frequency and its harmonics (red traces). For our systems, the breathing frequencies were in the range of 0.5–1 GHz for the $Si_3N_4$ microresonator (FSR ~100 GHz, $\kappa$~200 MHz) and 1–4 MHz for the $MgF_2$ platform (FSR ~14 GHz, $\kappa$~200 kHz). The breather regime can also be evidenced when measuring the repetition rate beatnote on the ESA. The oscillating pulse dynamics give rise to additional sidebands around the repetition rate, spaced by the breathing frequency (see Fig. 2f, h which compares stationary and breathing states in both platforms).

Another characteristic signature of the breathing state is observed in the optical spectrum. Figure 2b shows the measured spectra of both stationary and breathing single soliton based frequency combs, in a $MgF_2$ resonator. In the stationary state, the spectrum has a squared hyperbolic secant envelope corresponding to the stationary soliton solution, while in the breathing state, the spectrum features a triangular envelope (on a logarithmic scale), resulting from the averaging of the oscillating comb bandwidth by the optical spectrum analyser, as illustrated in Fig. 1c. The simulated spectrum (averaged over one breathing period) reproduces well the triangular feature. The weak sidebands on the optical spectrum are also captured on both the measured spectrum and averaged simulated spectrum (marked by arrows in 2b), providing an evidence for evanescent waves radiation by breather solitons.

**Breathing dissipative solitons dynamics**. Having established a deterministic access to breathers, we next characterised the breathing dynamics. We use a vector network analyser (VNA) to acquire the transfer function of the system from pump phase modulation to the transmitted power, which enables us to determine the effective laser detuning of the driven nonlinear system[28, 36]. In the stationary soliton state, this transfer function exhibits a double-resonance feature (red curve in Fig. 3c) which reflects the bistable nature of the intracavity field (soliton and CW background). The first one ($\mathcal{C}$-resonance) corresponds to

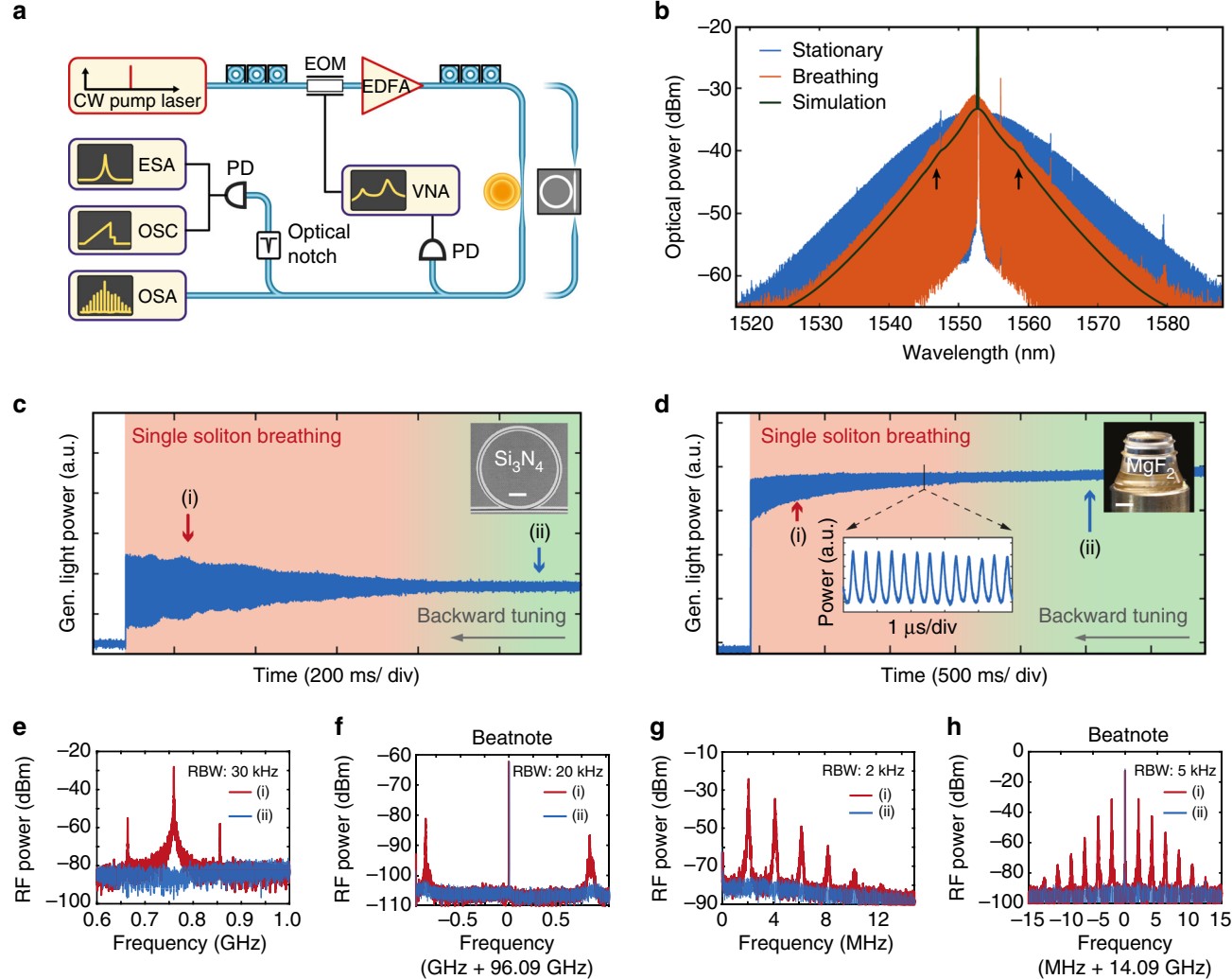

**Fig. 2** Experimental observation of breathing solitons in $Si_3N_4$ and $MgF_2$ platforms. **a** A similar experimental set-up is used for both platforms: A tunable continuous wave laser is used as a pump source. EDFA, erbium-doped fibre amplifier; EOM, electro-optical phase modulator; OSA, optical spectrum analyser; PD, photodiode; OSC, oscilloscope; ESA, electronic spectrum analyser; VNA, vector network analyser. **b** Experimental optical spectra of a stationary (*blue*) and breathing soliton states (*red*), in the 14 GHz FSR $MgF_2$ crystalline resonator. The effective detuning $\delta$ is varied by 0.5 MHz between the two states. The simulated optical spectrum averaged over one breathing period (*black line*) was offset by 3 dB for better visibility. The *arrows* mark the positions of weak sidebands visible in both the simulated and measured spectra. **c** Generated light power evolution for a single-soliton state in the 100 GHz $Si_3N_4$ microresonator as the pump is tuned backward, showing the transition from stationary state (*green shading*) to breathing (*red shading*) and final decay. The *inset* shows an SEM image of the microresonator used (the *scale bar* corresponds to 100 μm). **d** In the $MgF_2$ crystalline resonator (see inset, the *scale bar* corresponds to 2 mm), the comb light evolution features a similar behaviour as in **d**, when tuning backward. The *inset* shows the oscillations of the generated comb power, resolved with a fast photodiode and high sampling rate. **e**, **f** RF spectra of the generated light for a breathing (point (i) in **c**, **d**, *red trace*) and stationary (point (ii) in **c**, **d**, *blue trace*) soliton state respectively in the $Si_3N_4$ and $MgF_2$ resonators. In **f**, the 0.4 GHz span is centred at 0.8 GHz, close to the fundamental breathing frequency. The resolution bandwidth (RBW) is indicated on the corresponding panel. **g**, **h** Repetition rate beatnote for a breathing (i, *red*) and stationary (ii, *blue*) soliton state in the $Si_3N_4$ and $MgF_2$ resonators

the background cavity resonance, and indicates the effective pump laser detuning $\delta$ with good approximation. The second one ($\mathcal{S}$-resonance) corresponds to a resonant response of the soliton to the pump modulation. It emerges at lower frequency and is weakly dependent on the pump laser detuning.

Figure 3 shows the evolution of a single soliton in a $MgF_2$ resonator while tuning backward from the stationary state (pump power of 200 mW). During the scan, the transfer function of the system is monitored simultaneously with the comb repetition rate beatnote and total comb power. As the laser detuning is reduced, the $\mathcal{C}$-resonance consequently shifts to lower frequencies (Fig. 3a). Interestingly, both $\mathcal{C}$-resonances and $\mathcal{S}$-resonances are also

observed in the comb beatnote measurement, appearing as features on the background noise of the electronic spectrum analyser (dashed lines in Fig. 3b). We ascribe this effect to the transduction of laser input noise via the response of the system (i.e., incoherent response which is identical to the probed coherent response). The transition from stationary to breathing soliton occurs when the separation of the $\mathcal{C}$- and $\mathcal{S}$-resonances is comparable to the linewidth ($\kappa/2\pi$), for a detuning $\delta \sim 4$ MHz. Afterwards, in the breathing region, strong sidebands at the soliton breathing frequency and its harmonics emerge around the beatnote. The sidebands move progressively closer to the beatnote, revealing that the breathing frequency decreases for

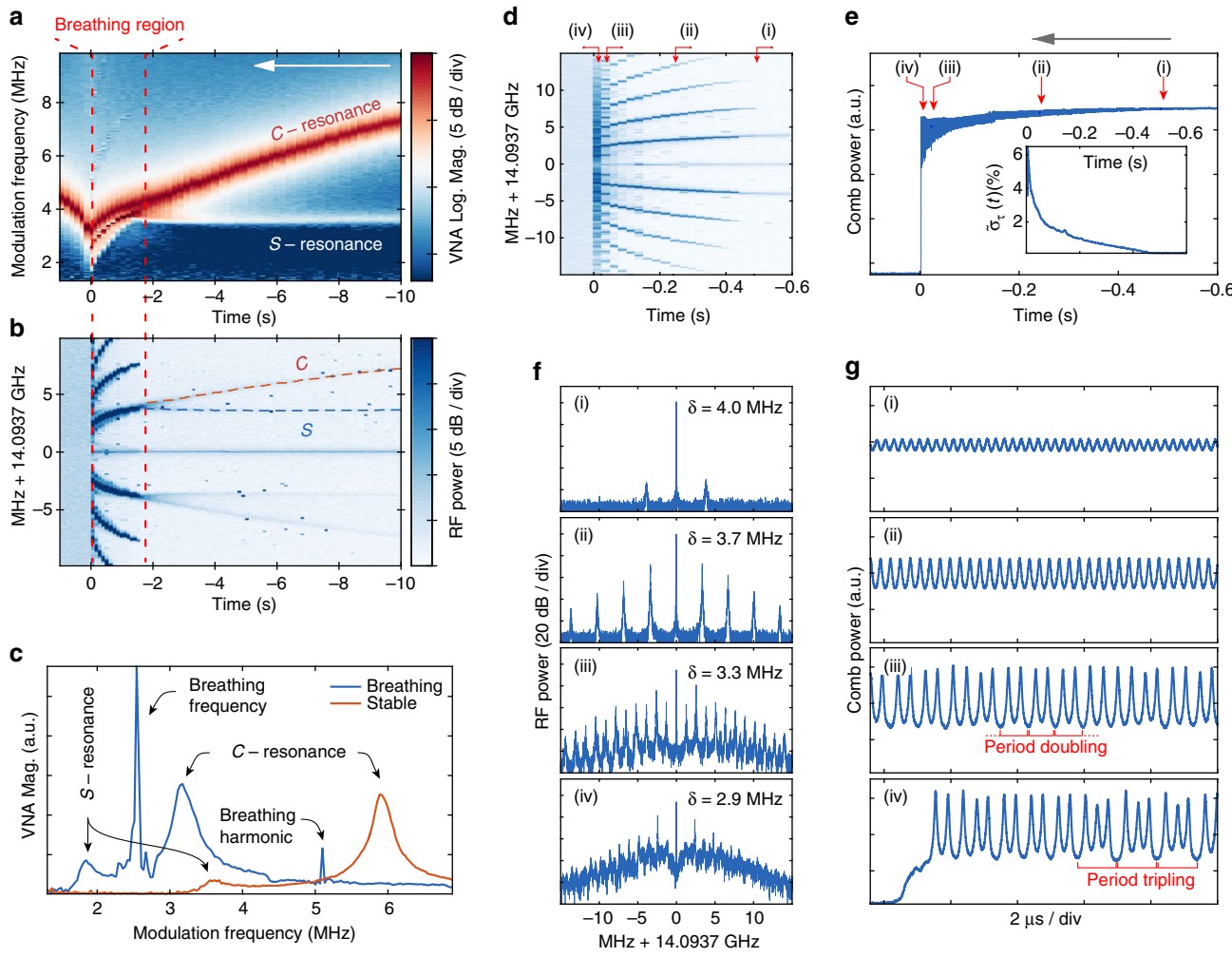

**Fig. 3** Evolution of the breathing dynamics of a single soliton in backward tuning in the MgF$_2$ resonator. **a** Map of concatenated vector network analyser (VNA) traces showing the evolution of the modulation response (log scale) from stationary soliton on the right of the time axis to breathing and decay on the *left* (the time origin $t = 0$ s, is set at the soliton decay). As the laser is tuned towards shorter wavelength, the effective detuning ($\mathcal{C}$-resonance) is reduced. The breathing starts typically when the separation of the $\mathcal{C}$- and $\mathcal{S}$-resonances is on the order of the resonator linewidth. **b** Corresponding spectrum of the comb repetition rate beatnote. The modulation response measured on the VNA is also visible in the noise of the RF beatnote spectrum (the *dotted lines* correspond to the $\mathcal{C}$-frequencies and $\mathcal{S}$-frequencies determined on the VNA). The breathing is indicated by the formation of sidebands around the repetition rate beatnote. As the detuning is reduced, the breathing frequency decreases until the soliton is lost. **c** Recorded modulation response in the state of a breathing and stationary soliton (linear scale). **d** Zoom-in spectrogram of the repetition rate beat within the breathing region and **e** corresponding generated comb light evolution. The *inset* shows the evolution of the relative standard deviation. **f** Repetition rate beatnote spectra in the various breathing stages (i)–(iv) highlighted in **d**, **e** (resolution bandwidth 2 kHz). **g** Recording of generated light power oscillations at the points (i)–(iv) highlighted in **d**, **e**

smaller detuning. In the breathing state, the transfer function (blue curve in Fig. 3c) features a strong sharp peak at the breathing frequency that appears in between the $\mathcal{C}$-resonance and the $\mathcal{S}$-resonance. From this response, the breathing frequency and the effective laser detuning can thus be measured with a good precision. Notably, the $\mathcal{S}$-resonance behaviour is greatly modified in the breathing domain as it shifts together with the breathing frequency and detuning. We also observed that the transition into the breather regime is reversible by tuning the laser forward (back into the stationary state).

Figure 3d–g shows the detailed breathing dynamics within the breathing region. In particular, the comb power is measured in two ways. First, the global evolution is monitored continuously on a DC-coupled photodiode with a slow sampling rate of ~100 kSa/s (Fig. 3e). Thus, the faster breathing oscillations appear as increased amplitude noise, which can be quantified with the relative standard deviation $\tilde{\sigma}_\tau(t) = \sigma_\tau(t)/\mu_\tau(t)$, where $\sigma_\tau$ and $\mu_\tau$

are the local standard deviation and mean power level over $\tau = 1000$ samples. Second, the fast dynamics of the intracavity soliton is also recorded on a real-time oscilloscope with 120 GSa/s, but in short sequences spread over the scan. This allows the observation of the pulsed nature of the intracavity pattern (see Fig. 5). The breathing oscillations in each sequence are then recovered by detecting the envelope of the resolved pulse train and down-sampled (Fig. 3g). Ultimately, the breathing dynamics could be resolved with a slower oscilloscope.

The breathing starts with a weak oscillation of the soliton pulse train power (stage i, $\delta \sim 4$ MHz). This corresponds to a single pair of weak sidebands on the comb beatnote. For smaller detuning the breathing becomes stronger, so that the first sidebands (fundamental breathing frequency) increase, and breathing harmonics emerge (stage ii) as the breathing pattern is not sinusoidal. At $\delta \sim 3.3$ MHz (stage iii) the system exhibits a period doubling, which corresponds to the appearance of sub-sidebands

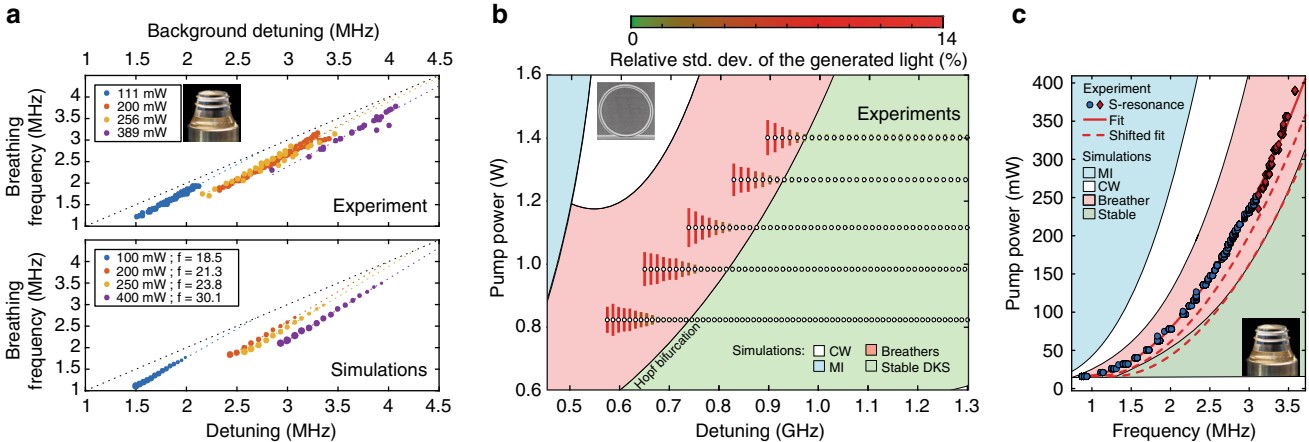

**Fig. 4** Breathing dynamics and dependence on the parameters of the system. **a** *Top*: Experimental determination of the breathing frequency evolution with the detuning for different pump powers, retrieved by the modulation response measurement. *Bottom*: Simulated evolution of the breathing frequency (see Methods section for details). **b** Stability chart of the CW-pumped $Si_3N_4$ microresonator in pump power over effective detuning coordinates. *White filled circles* indicate experimentally accessed DKS states. The colour-coded vertical lines indicate the measured relative standard deviation of the generated light power. The coloured background regions and boundaries are interpolated from simulation results (see Methods section for details) and correspond to: CW-state (*white*)—the soliton decays to the homogeneous background, chaotic modulation instability (*blue*), stationary DKS state (*green*) and breathing DKS state (*red*). **c** Dependence of the $\mathcal{S}$-resonance on the pump power, measured for a stationary soliton. This resonance provides an estimate of the detuning point at which the breathing starts. The measurements were carried with the detuning stabilised to $\delta = 3.5$ MHz for $P < 300$ mW (*blue dots*) and $\delta = 7$ MHz $P > 300$ mW (*red marks*). The evolution fits to a parabolic dependence. The regions were retrieved with a polynomial fitting of the boundaries from a simulated stability chart (see Methods section for details). The Hopf boundary obtained from the simulations is contained within a $[+1, +2]\kappa/2\pi$ margin (*dashed lines*) to the measured $\mathcal{S}$-frequency, which is in agreement with the experimental observations reported in Fig. 3a

with frequency half of the initial breathing frequency. At last, the breathing turns into strong and irregular oscillations (stage iv, $\delta \sim$ 2.9 MHz), exhibiting sporadic transitions to period tripling. This coincides with a large increase in the noise pedestal around the beatnote, although the fundamental breathing frequency remains distinguishable. Finally, the soliton decays quickly thereafter. Other sections of the same trace are presented in the Methods section, revealing further transitions to period 5, 7 and 8 oscillations within a very irregular breathing state. Such transitions to higher periodicity, temporal chaos, as well as the soliton collapse match the predicted evolution from numerical studies of the LLE[8, 18].

The combined effect of increased modulation depth and reduced breathing frequency is reminiscent of a typical characteristic of complex dynamical systems approaching critical transitions[37, 38]. In the present case, the critical event consists in the loss of the single soliton, either via collapse into the continuous background as observed in this work (switching), or via spatiotemporal instabilities[35]. When approaching a tipping point, early warning signals in the form of a variance increase and a critical slowing down have been reported in a wide variety of systems, ranging from lasers near threshold to entire ecosystems and the climate[39–41].

We next study the breathing frequency as a function of laser detuning for a single-soliton state in the $MgF_2$ resonator. The backward tuning over the breathing region was repeated for different pump power levels, and the breathing frequency was measured as a function of effective laser detuning (Fig. 4a) using the transfer function of the system. The detuning dependence is close to linear: $f_b \approx 1.23\delta + f_b^0$, where $f_b$ is the breathing frequency and $\delta$ indicates the effective laser detuning. The offset $f_b^0$ is observed to decrease with the pump power. We performed numerical simulations based on the LLE and obtained an excellent quantitative agreement, with almost identical results (Fig. 4a). A direct linear relation between the breathing frequency and the detuning is also suggested by the approximate breather expression we derived analytically (see Methods section).

The relation we measured is of opposite sign to a trend reported by Yu et al.[20], where the breathing frequency was shown to increase for reduced detuning. Although numerical simulations predict an inversion of the trend over a narrow range of detuning, for small pump amplitudes and detuning (see Methods section), the dominant evolution of the breathing frequency matches with our observations. In ref.[20], the detuning was not accessed directly, and the measurements carried with multiple solitons in the cavity, making any comparison difficult.

**Breathing region**. We experimentally studied and mapped the stability chart of DKS solitons in the two-parameter space (pump power $P_{in}$ and effective detuning $\delta$) of the CW-pumped micro-resonator system[9, 42, 43]. A stationary single soliton was generated using the backward tuning method at different pump powers, and gradually tuned across the breathing region until its decay. The white circles in Fig. 4b mark the operating points ($P_{in}$, $\delta$) thus accessed experimentally. The colour-coded vertical line around each circle indicates the relative standard deviation of the output power measured at the corresponding point and directly relates to the breathing amplitude. The results reveal a pump power dependency for the breathing threshold, whose location shifts towards higher effective detuning values and range slightly reduces when the pump power increases, as predicted in ref.[18].

We compared our experimental results to LLE-based simulations (see Methods section for details). The resulting types of intracavity field attractors at various operating points are labelled via the background colour-coding in Fig. 4b: CW-state (white), where the soliton decays to the homogeneous background; chaotic modulation instability state (blue); stationary soliton state (green); and breathing soliton state (red). The experimentally accessed stationary and breathing states are well within the corresponding areas predicted by the simulations. The deviation between experimental results and simulations for the transition boundary (Hopf) can be attributed to the deviations between the measured detuning values and the true $\delta$ that differ at

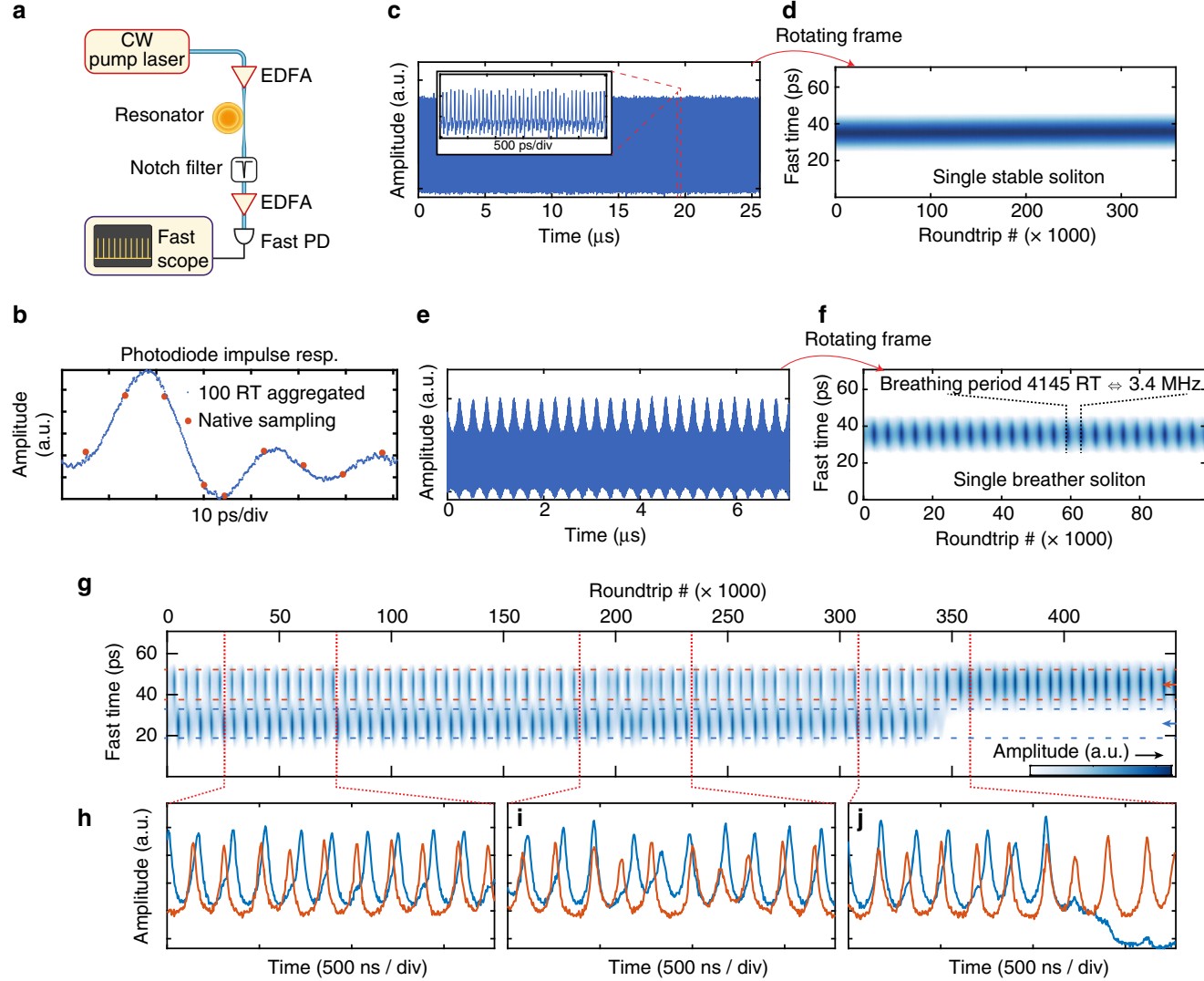

**Fig. 5** Direct observation of the spatiotemporal dynamics. **a** Experimental set-up. EDFA, erbium-doped fibre amplifier; PD, photodiode. **b** Photodiode response. The red dots mark the original sampling over a single roundtrip period (RT). With 9 points per period, the pulse amplitude cannot be accurately resolved. This problem is solved by aggregating 100 roundtrips to increase the effective sampling rate and retrieve the impulse response to a single soliton. **c** Single soliton pulse train, containing $3.5 \times 10^5$ roundtrips. The *inset* shows a short section of the trace, where individual pulses can be coarsely located. **d** Dividing the trace into groups of 100 aggregated roundtrips, and stacking reveals the spatiotemporal evolution of the soliton. The soliton position and amplitude is fixed as the soliton is stable. In this map, the colourmap is set to remove the ripples of the photodiode response. **e** Single breathing soliton pulse train. **f** Applying the same procedure as in **d** reveals the oscillating pulse amplitude while its position remains stable. **g** Spatiotemporal evolution of a breathing two-soliton state undergoing a transition to a breathing single-soliton state (switching). **h–j** show the evolution of the amplitude of each soliton. **h** Traces showing a $\pi/2$ phase difference between the breathing oscillations of the solitons. **i** Unstable breathing, after which the quadrature relation is restored. **j** Collapse of one soliton, while the other survives and remains in the breathing region

low-detuning due to the higher background[28]. The highly unstable and short-lived breather in this region makes it harder to resolve. Finally, high-order dispersion and nonlinear effects (e.g., Raman scattering, avoided mode crossings and third order dispersion) were not included in the simulations for simplicity, but are present in the $Si_3N_4$ microresonator.

Furthermore, as noted earlier, the breathing emerges when the $\mathcal{C}$-resonance is tuned close to the $\mathcal{S}$-resonance, and their separation is on the order of the resonator linewidth. Therefore the $\mathcal{S}$-resonance frequency provides an estimate for the detuning value of the upper boundary of the breathing region (Hopf bifurcation). Experimentally, we monitor the $\mathcal{S}$-resonance frequency as the pump power is raised, while stabilising the laser detuning ($\mathcal{C}$-resonance frequency) to a constant value in the

stationary soliton state[26]. Figure 4c reports the evolution of the $\mathcal{S}$-resonance frequency with the pump power for the $MgF_2$ resonator, whose smaller linewidth produces narrower resonance peaks in the transfer function that are easily resolved. The obtained relation fits to a parabolic dependence and matches the Hopf boundary retrieved from simulations with a frequency offset that does not exceed twice the linewidth, showing that the breathing region can be identified even from the stationary state.

**Real-time observation of breathers**. The fast soliton dynamics in the microresonator is studied further in the time domain by measuring the 14 GHz soliton pulse train coupled out of the $MgF_2$ resonator. The generated light is amplified and detected on a fast

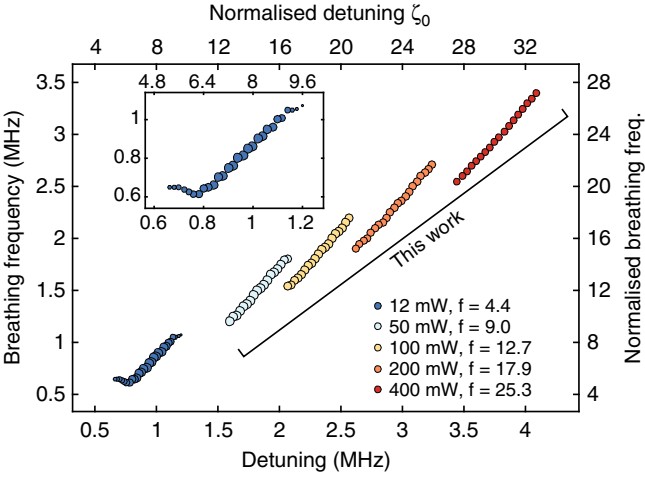

**Fig. 6** Simulated evolution of the breathing frequency as a function of the detuning and pump power. The breathing frequency is normalised to the half width at half maximum of the cavity ($2f_b/\kappa$). In these simulations, we assumed $\kappa/2\pi = 250$ kHz

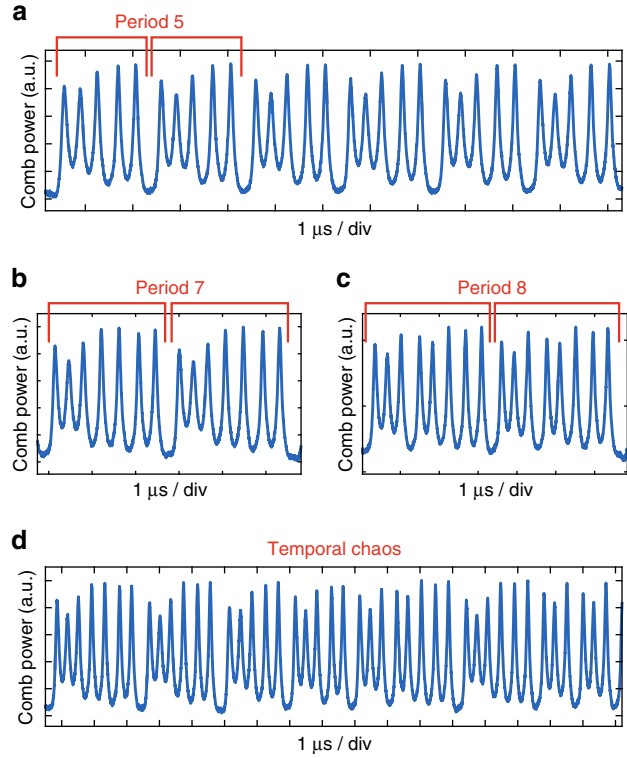

**Fig. 7** Experimental observation of high periodicity breathing and temporal chaos. The periodicity is assessed when the pattern is repeated at least twice. **a** Period 5 oscillations **b** Period 7 oscillations **c** Period 8 oscillations **d** Irregular breathing, no clear periodicity can be determined

photodiode (70 GHz bandwidth) connected to a real-time oscilloscope with 45 GHz analogue bandwidth (sampling rate 120 GSa/s). We note that so far, the real time sampling of successive solitons in microresonators had not been attained due to the required high sampling bandwidth. The present configuration allows for the measurement of ~9 samples per roundtrip and enables a coarse localisation of the soliton pulse within one roundtrip as shown in Fig. 5b, c. Since we observe that the soliton breathing dynamics evolve over a large number of roundtrips (>1000), we aggregate together the samples contained in segments of 100 roundtrips, to achieve an effectively larger sampling rate. This produces smoother traces, revealing the impulse response of the acquisition system (matching with the photodiode response), where the instantaneous pulse peak amplitude can be reliably retrieved (Fig. 5b). Longer traces (Fig. 5c, e) that measure the evolution over a large number of roundtrips are divided in 100-roundtrips segments, which are aggregated and stacked. This facilitates the visualisation of a spatiotemporal evolution of the cavity content over a large number of roundtrips.

We first benchmarked our measurement procedure in the single-soliton state. At a pump power of 230 mW and for the effective laser detuning ~10 MHz, the soliton is stationary as expected, with a constant amplitude (Fig. 5d). For a smaller detuning ~3.5 MHz, the soliton is breathing and the time trace reveals the oscillatory envelope of the soliton amplitude (Fig. 5e). In the spatiotemporal frame, this leaves a dotted pattern at the breathing period (Fig. 5f), where the blue shading indicates the soliton amplitude. The breathing frequency is ~3.4 MHz corresponding to 4145 roundtrips.

The fast recording on the real-time oscilloscope also enables us to delineate the breathing dynamics of individual pulses in a multiple-soliton state. Figure 5g shows the evolution of a breathing two-solitons state during a backward tuning around $\delta \sim 2.1$ MHz. The state experiences a switching[28] where one soliton decays and the other survives. Furthermore, in this small detuning condition, the breathing is typically irregular and might be locally identified as period doubling or tripling, as reflected on the traces (Fig. 5g–j). The measurement reveals that the two solitons breathe overall at the same frequency but are not in phase. In the present case, there seems to exist a preferred phase relation of ~$\pi/2$. Even if the breathing is irregular and the phase relation can be locally altered as shown in Fig. 5i, the relative phases seem to quickly recover this relation. A longer section of

the spatiotemporal evolution of Fig. 5g can be visualised in the Supplementary Movie 1, together with a second similar dual-breather realisation (Supplementary Movie 2). Such behaviour has been predicted by Turaev et al.[34], showing that the longer interaction length of breathing solitons can lead them to form bound states with a specific separation distance and breathing phase relation. A quadrature breathing should correspond to a comparatively large soliton separation, which matches with the above case as the pulses are separated by more than the photodiode response time. However, we could not derive a clear correlation between the soliton separation and the relative breathing phase. Other cases of in-phase, out-of-phase, and quadrature breathing, as well as phase wrapping were also observed and are reported in Methods section.

## Discussion

We have experimentally demonstrated the formation of breathing dissipative solitons in two distinct microresonator platforms. The large difference in the characteristics of the $MgF_2$ crystalline resonator and photonic chip $Si_3N_4$ microresonator validates the universal nature of our observations. We implemented a laser tuning method which enables a reliable access to soliton breathing. Typical signatures of breathing solitons, including a periodically varying soliton peak intensity and a triangular spectral envelope are identified and observed. Moreover, we presented a direct time-resolved observation of dissipative Kerr solitons in microresonators, revealing the breathing dynamics of individual solitons in both single and multiple breathing soliton states. Such measurements unambiguously reveal the transition to higher breathing periodicity and a more chaotic type of behaviour. By monitoring the laser detuning of the driven nonlinear system, we present direct measurements of the breathing

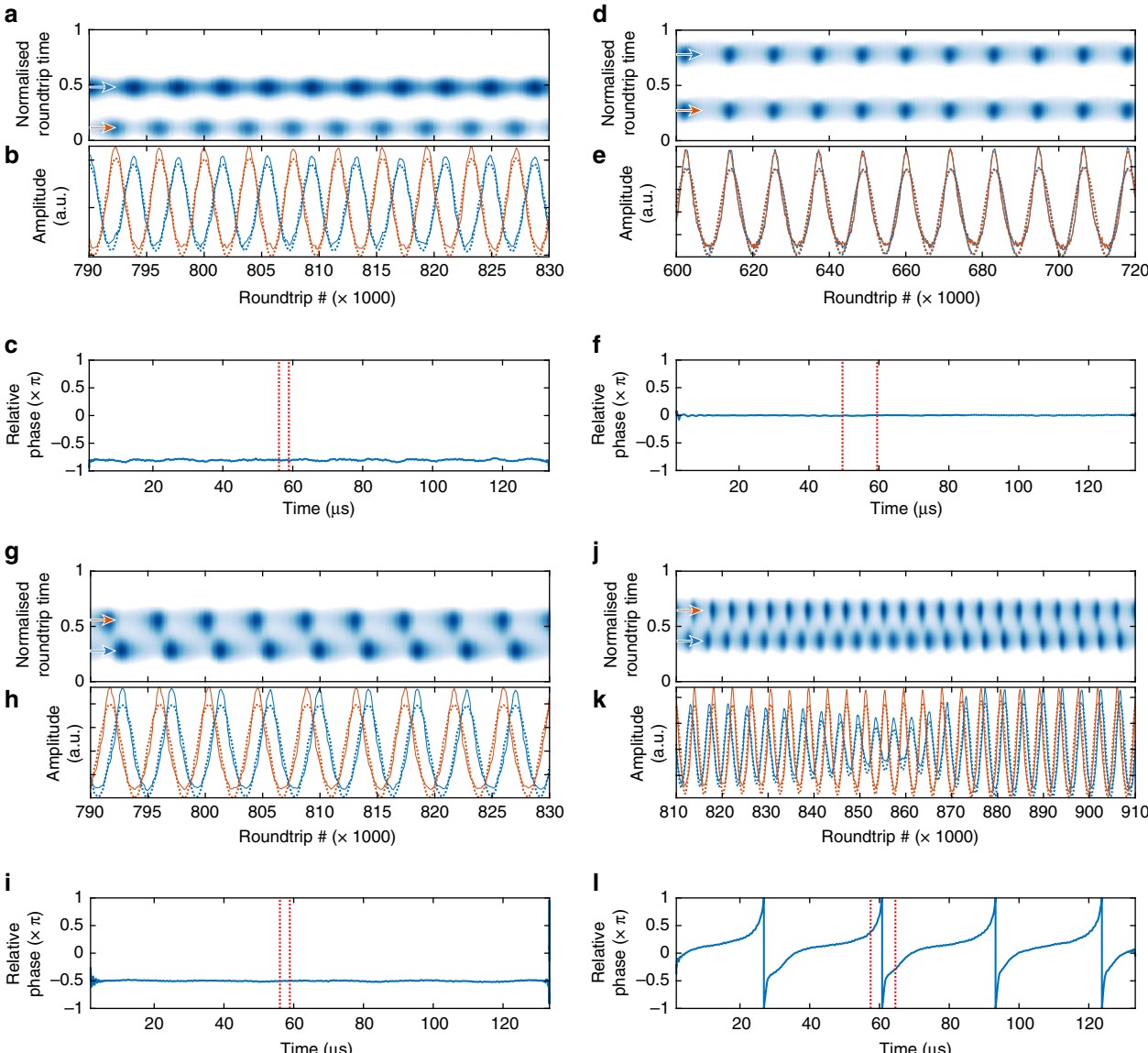

**Fig. 8** Relative breathing phase. Four realisations are presented. **a**, **d**, **g**, **j** Expose the spatiotemporal map of a fraction of the trace. The *blue* and *red curves* of **b**, **e**, **h**, **k** display the evolution of the amplitude of each soliton. The plain (*dashed*) lines show the raw (*filtered*) signals. **c**, **f**, **i**, **l** Show the evolution of the relative breathing phase across the entire trace. The *dashed red interval marks* the section of the trace displayed in the two panel above. Upon various two breathing soliton state realisations, the relative breathing phase is observed to vary. **a–c** Out-of-phase breathing. **d–f** In-phase breathing. **g–i** Quadrature breathing. **j–l** At the breathing onset, the phase continuously accumulates but the breathing tends to synchronise when close to in phase

frequency dependence on the laser detuning. These studies evidenced a linear relation, which agrees remarkably well with numerical simulations and provides further insights into this breathing property. Furthermore, the experimental mapping of the Hopf transition boundary from stationary to breathing state reveals the parabolic-like relation between the pump power and the detuning, which also matches numerical simulations. In the context of low-noise operation of soliton-based Kerr frequency combs, breathing degrades the soliton stability and should generally be avoided. Our results provide useful diagnosis tools to determine the breathing boundary, even from the stationary soliton state and shine a new light on the $\mathcal{S}$-resonance of the soliton and its relation to the Hopf boundary and breathing frequency. These findings not only carry importance from an application perspective, but also contribute more broadly to the fundamental understanding of dissipative soliton physics. Our observations further demonstrate the suitability of the

microresonator platform to study nonlinear dynamics, especially for accessing high normalised driving amplitudes. In the present case, the remarkable agreement between the numerical simulations and experimental observations validates one more time the relevance of the Lugiato-Lefever model, even in such cases of nonstationary and chaotic dynamics. However, beyond the intrinsic soliton breathing explored in this work (as predicted by the standard LLE) the imperfections of a microresonator provide an additional opportunity to study the effects of perturbations on soliton stability, such as the influence of mode-family interactions triggering inter-mode breathing of solitons[44].

## Methods

**Optical resonators**. $Si_3N_4$ integrated microring resonators with the free spectral range (FSR) of ~100 GHz and $Q$-factors ~$10^6$ (linewidth $\frac{\kappa}{2\pi} = 150 - 200$ MHz) was fabricated using the photonic damascene process[45]. In order to achieve the single-mode operation and suppress the effect of avoided mode crossings, a filtering

section was added to the microresonator[46, 47]. The resonators dispersion parameters—$-\frac{D_2}{2\pi} = 2\,MHz$, $\frac{D_3}{2\pi} = \mathcal{O}(1\,kHz)$—were measured using the frequency comb assisted laser spectroscopy method[48] (the resonance frequencies near $\omega_0$ are expressed in a series $\omega_\mu = \omega_0 + \sum_{i \geq 1} D_i \mu^i / i!$, where $i \in \mathbb{N}$, $\mu \in \mathbb{N}$ is the mode number). The wavelength of CW pump laser in experiments was set at 1553 nm, the pump power varied from 1 to 4 W.

The MgF$_2$ crystalline resonator with FSR $\frac{D_1}{2\pi} = 14.094\,GHz$ was fabricated by diamond turning of a cylindrical blank. The high Q-factor of ~$10^9$ (intrinsic linewidth $\frac{\kappa_0}{2\pi} = 80\,kHz$, intrinsic finesse $\mathcal{F} \sim 1.7 \times 10^5$) was achieved with subsequent hand polishing. The dispersion parameters at the pump wavelength of 1553 nm are: $\frac{D_2}{2\pi} = 1.96\,kHz$, $\frac{D_3}{2\pi} = \mathcal{O}(1\,Hz)$. The pump laser (fibre laser, wavelength 1553 nm; short-term linewidth 10 kHz) is amplified between ~20 and 450 mW and evanescently coupled to the resonator with a tapered optical fibre, which enables a tuning of the coupling. The loaded linewidth $\kappa$ is retrieved by measuring the $\mathcal{S}$-resonance linewidth in the VNA trace (when no solitons are present in the cavity), the associated coupling coefficient $\eta = \kappa_{ex}/\kappa = (\kappa - \kappa_0)/\kappa$ was measured in the range 0.45–0.62.

**Numerical simulations**. Numerical simulation based on the LLE (Eq. 1) were implemented in order to study breathers. For the MgF$_2$ resonator, the simulations are performed using periodic boundary conditions with 1024 discretisation points (1024 modes). The simulation of the breathing frequency as a function of the control parameters is carried as follows: the operating parameters (pump power and the laser detuning) are fixed and the intracavity field is initiated with a single (stationary) soliton ansatz[2]. The simulation of the soliton evolution is then carried over 15 photon lifetimes ($2\pi/\kappa$) and the breathing dynamics analysis is carried over the final 2/3 time range, where the stationary soliton ansatz has converged to the inherent breathing state of the system. The oscillation frequency is determined via spectral analysis and plotted in Fig. 4a. We also simulated the breathing frequency in the range of parameters simulated in ref. [20] and compared with our operating conditions on Fig. 6. The inversion of the slope is reproduced at low pump amplitude and small detuning, but occurs over a very narrow range of the parameter space (on the order of the resonator linewidth), which we believe is difficult to access experimentally. The dominant trend is well reflected in our measurements as shown in Fig. 4a.

The simulations of the stability chart of the Si$_3$N$_4$ microresonator presented in Fig. 4b were performed with 512 modes. Using hard excitation scheme, stationary DKS were seeded at fixed input powers and large detunings. Then the laser detuning was reduced step by step to map over the chart. In each step, the intracavity field pattern is characterised after ~5000-roundtrips to exclude early-stage transient formations. In simulations for both MgF$_2$ and Si$_3$N$_4$ microresonators, we identified intracavity states of single stationary soliton state, breathing soliton state, chaotic state in the operation regime of modulation instability (MI) and state where intracavity field decays leaving only the cw background, showing in colour-codings in Fig. 4b, c.

**Approximate breather ansatz**. We develop an approximate breather solution for the LLE (Eq. (1)), that allows to inspect the relation of the breathing regime parameters to the pump power and the effective detuning. It is known that an approximate stationary solution of the LLE for positive $\zeta_0$ (i.e., for the pump laser being effectively red detuned) may be found as a sum of the soliton and a background:

$$\Psi(\theta) \approx \Psi_C + \Psi_S(\theta)\, e^{i\phi_0}, \tag{3}$$

here $\Psi(\theta)$ is the intracavity waveform, $\theta = \phi\sqrt{\frac{1}{2d_2}}$ is the dimensionless longitudinal coordinate, $\phi$ is the co-rotating angular coordinate of the resonator and $\bar{d}_2 = D_2/\kappa$ is the dimensionless dispersion. $\Psi_C \approx -if/\zeta_0$ represents the constant solution of (1) (background), while $\Psi_S = B\,\text{sech}(B\theta)$ is the exact stationary conservative soliton solution of (1) (without loss or drive), with $B = \sqrt{2\zeta_0}$. The phase $\phi_0$ may be found by perturbation methods[49] from $\cos\phi_0 = 2B/\pi f$.

The exact Kuznetsov-Ma breather[21, 22] solution of Eq. (1) without loss and pump can be employed to derive an approximate ansatz for dissipative breathing solitons:

$$
\begin{aligned}
\Psi_S(\theta, \tau) &= \left( \frac{K_1 \cos\Omega\tau + iK_2 \sin\Omega\tau}{\cosh B\theta - K_3 \cos\Omega\tau} - \epsilon \right) e^{iK_4\tau} \\
\Omega &= \frac{B}{2}\sqrt{B^2 + 4\epsilon^2} \\
K_1 &= \frac{B^2}{\sqrt{B^2 + 4\epsilon^2}}, \quad K_2 = B \\
K_3 &= \frac{2\epsilon}{\sqrt{B^2 + 4\epsilon^2}}, \quad K_4 = \epsilon^2 - \zeta_0
\end{aligned}
\tag{4}
$$

If the time dependent part of the background $\epsilon$ is small then leaving only terms up to the first order on $\epsilon \to 0$ we arrive at:

$$\Psi_S(\theta, \tau) = B\,\text{sech}(B\theta) + 2\epsilon\cos(\zeta_0 t)\,\text{sech}^2(B\theta) - \epsilon\, e^{-i\zeta_0 t}. \tag{5}$$

We notice, that for $\epsilon = 0$ this breather converges to a simple stationary soliton, and

for small $\epsilon$ the oscillation frequency of both the background and soliton itself simply coincides with the laser detuning.

**Higher breathing periodicity**. At low detuning, the breathing gets irregular, as shown in Fig. 3g. Within this regime of irregular oscillations, we observed episodic transitions to high periodicity breathing, where the breathing pattern repeats at least twice. Apart from the case of period tripling shown in the main manuscript, we present in Fig. 7 the observed transitions to period 5, 7 and 8 oscillations. A representative section of the trace showing irregular (chaotic) breathing, where no clear periodicity can be found, is also displayed.

**Relative breathing phase**. We have recorded the spatiotemporal evolution of several dual breathers realisations and analysed the relative breathing phase (Fig. 8). The detuning is overall larger than the case shown in Fig. 5g, so that the breathing is more regular. The breathing synchronisation is assessed in the following way: a cut of the spatiotemporal map is extracted along the peak corresponding to the position each soliton. The two resulting time series are band-pass filtered to keep only the fundamental breathing harmonic and the analytical signal of each filtered trace is computed via Hilbert transform. The relative phase is detected by taking the argument of the quotient of the two analytical signal.

Typical breathing behaviour are displayed in Fig. 8a–c, showing out-of-phase, in phase and quadrature breathing. Besides these three cases, other phenomenon were observed punctually such as synchronisation with a phase multiple of $\pi/4$, of incomplete synchronisation at the onset of breathing as shown in Fig. 8d. The spatiotemporal evolution of Fig. 8d can be visualised in Supplementary Movie 3.

**Data availablity**. The code and data used to produce the plots within this paper are available at 10.5281/zenodo.823538. 545822. All other data used in this study are available from the corresponding authors upon reasonable request.

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

## Acknowledgements

The authors gratefully acknowledge V. Brasch, for his help in setting up the fast detection scheme, as well as the valuable discussions and support of J.D. Jost, M.H.P. Pfeiffer, M. Anderson and M. Geiselmann. This publication was supported by the Swiss National Science Foundation (SNF) under grant agreement 161573 as well as Contract W31P4Q-14-C-0050 (PULSE) from the Defense Advanced Research Projects Agency (DARPA), Defense Sciences Office (DSO), and by the European Space Agency (ESA), European Space Research and Technology Centre (ESTEC), Contract No. 4000116145/16/NL/MH/GM. This material is based upon work supported by the Air Force Office of Scientific Research, Air Force Material Command, USAF under Award No. FA9550-15-1-0099. M.K. acknowledges funding support from EU FP7 programme under Marie Sklodowska-Curie ITN grant agreement No. 607493. H.G. acknowledges funding support from EU Horizon 2020 research and innovation programme under Marie Sklodowska-Curie grant agreement No 709249. M.L.G. acknowledges support from the Russian Science Foundation grant #17-12-01413 and Russian Foundation for Basic Research grant #17-02-00522. All samples were fabricated and grown in the Centre of Micro-NanoTechnology (CMi) at EPFL.

## Author contributions

E.L. and M.K. designed and performed the experiments respectively in $MgF_2$ and $Si_3N_4$ resonators, analysed the data, and carried out the simulations with input from H.G. and M.L.G. M.L.G. developed the analytical model and helped in the data analysis. All authors discussed the data and contributed to the manuscript. T.J.K. supervised the project.

## Additional information

**Competing interests:** The authors declare no competing financial interests.

