## [Peer Review File · Nature Communications]

Reviewers' comments:

Reviewer #1 (Remarks to the Author):

This paper reports a study of the dynamics of breathing (oscillating) solitons in two different platforms.

The study is extremely accurate and it provides a complete picture of the dependence of the breathers' stability and frequency in the relevant parameter space made of cavity detuning and pump power. The agreement with numerical simulations based on the LLE is very good. Remarkably, the paper also includes a real time analysis of breathers' dynamics.

Although similar studies appeared recently, one some months ago (Ref. 20) and the other just some weeks

ago (Ref. 21), I think that the quality of the research presented here is not diminished by those previous

publications, and the paper deserves publication on Nature Communications.

However, the publication in particular of [21], which occurred after the submission of the present paper, imposes some changes.

First, the claim of priority in lines 273-274 must be removed.

Second, a critical comparison with the results of [21] is needed. I found an apparent contradiction between Fig.

3a of the present manuscript and Fig. 5 of [21]. Both figures show a linear dependence of the breathing frequency

on the effective cavity detuning, but the slopes have opposite signs. Yet, I believe that the contradiction is

only apparent, because if one looks at the supplementary material of [21] one realizes that Fig. 5 is just a piece

of a larger picture, where for large detuning the breathing frequency increases with the detuning as in Fig. 3a.

Since the detuning is measured in units of the cavity linewidth, which is very small in the microresonator

considered in Fig. 3a, I guess that the detuning is really very large in that figure.

Anyway, a comment on this point would be welcome.

On the other hand, the two figures agree in that they show a shift of the breathers toward larger detuning

for increasing power.

Apart from this I have very few remarks, because the paper is really well written and the results are presented

in a very clear way.

Just few points:

- the reference to Fermi-Pasta-Ulam recurrence in the abstract is in my opinion not appropriate, because

that feature is not investigated in the paper. The better place for mentioning FPU is the introduction

- the comment about critical slowing down at lines 181-185 must be either substantiated or removed, because

actually I do not see any "criticality" in the observed dynamics

- in "Methods" it would be useful to write the modal equations which are effectively used in the simulations

The quality of the figures is very good, but sometimes it is difficult to follow the order of the various

subfigures, because they are so many and they are not commented in the text in alphabetic order.

For instance Fig.

1b is commented after Figs. 1i.

Moreover, the global caption of Fig. 1 is "Experimental observation of ..." but Fig. 1a is a simulation.

Reviewer #2 (Remarks to the Author):

Breathing dissipative solitons in optical microresonators, Lucas, Karpov, Guo, Gorodetski, Kippenberg

The manuscript presents detailed observations of breathing solitons in microresonators. In particular, the breathing frequency and the region of existence of breathers is measured accurately as a function of both detuning and pump power. The dynamics is also resolved in the time domain.

The technical level of the paper is quite high and the results are clear. I would question however the suitability of the manuscript for Nature Communications. Two previous groups have now published studies of breathers in microresonators, respectively in Phys. Rev. Lett. (Ref 20) and in Nature Communications (Ref 21). Plus breathers have also been observed before in equivalent fiber resonators, as mentioned in the manuscript (Ref 19). I could readily agree that the current manuscript presents a more systematic, maybe more detailed study. Nature Communications is however targeted at "important advances of significance". In light of the other previous studies, this manuscript looks incremental. Observing breathers in microresonators, or even in a Kerr resonator, is not an important advance anymore. The novelty is thus weak. The authors vaguely mention that "a variety of open questions ... and dynamics remain unexplored" (or in the abstract, "lack a comprehensive understanding") but they do not really argue about what is different, novel, or significant in their work in comparison to the two other reported observations of breathers. For all these reasons, I think this paper is more suited for a more specialized journal.

Also, I do not think this paper brings more *understanding* (using the working of the abstract) to the breathers. Sure, their parameters are measured accurately and systematically, but no new insights (understanding) is actually brought forward in terms of why they occur, why does the breathing frequency changes the way it does, ...

Note: Ref 21 is cited as an arXiv. It is now published in Nature Comm 8, 14569 (2017) and the reference should be updated.

Having said that, there is no doubt that the work is valuable. I therefore provide below a list of more specific points that the authors should address when resubmitting to another journal.

1) Both in the abstract and at the beginning of the first paragraph, the authors try to point out the generality of dissipative solitons (in sand dunes, plasmas, ...) yet their definition are very much centered around temporal effect in optics. In the abstract (line 7-8) -> balance of *dispersion* and nonlinearity. Some dissipative solitons (outside of optics) are based on diffraction or even diffusion, not necessarily dispersion. In the 1st paragraph (line 24) -> "dissipative solitons are solutions of the damped NLSE", citing Ref 1. Ref 1 actually explores many systems, not all described by the NLSE, or damped NLSE, that exhibit dissipative solitons. This should be written more carefully. Also, the first sentence of the abstract ending with "ranging from sand dunes, plasma physics, to pulse formation ..." (line 9) does not sound grammatically correct to me. Only one item should be mentioned between "ranging from" and "to".

2) Line 28: "optical frequency combs with large repetition rates in the microwave domain": this is slightly misleading. It could be interpreted as "combs in the microwave domain", whereas the

combs considered here are in the optical domain. This could be clarified.

3) Lines 32-33: "microresonators appeared as a suitable platform to study DKS properties and dynamics". I wonder why? In some respect, observations of these solitons are much simpler/easier to perform in fiber cavities, where the dynamics are identical but occur over longer timescales (easier to resolve), and several papers have proven that point. In fact, in terms of exciting/observing these solitons, everything is harder in a microresonator.

4) Line 34, "field patterns" is ambiguous. This could be interpreted as meaning higher-order spatial modes.

5) Line 35, it is my understanding that "dark pulses" and "platicons" are synonymous, while here these are presented as different structures.

6) Line 51, "Understanding the breather regime ... [is] a necessity for applications" -> the only thing that appears necessary is to know where it is, to avoid it. The properties of the breathers per se are not important in the context of envisioned applications.

7) Line 59, the authors claim they "reveal a link between breathing frequency and driving laser parameters". The existence of that dependence (or link) is not a novelty. Other works on breathers have revealed that. Therefore it should not be presented as such a grand claim.

8) Top of page 4, the parameter "f" is referred to as "normalized power". It is however proportional to pump *amplitude* (square root of power). Therefore, I am not sure that referring to it as a normalized *power* is appropriate. Maybe normalized pump amplitude?

9) I was somewhat confused by the references to figures 1f,h and 1g,i. I am not sure what is the difference. 1f,h are presented as "low frequency RF spectra". While 1h starts at zero frequency (low frequency?), 1f starts at 0.6 GHz, which is not so low. Also, what is the difference between f and g (or h and i)? One is the RF spectra and the other is the beatnote, but I am not sure about the exact difference (see also lines 123-126). Are these measured differently? This should be explained better.

10) Line 120-122: still with regards to Fig 1, the authors talk about peaks revealing the *harmonics* of the breathing frequency, yet no such harmonics are visible for the SiN platform.

11) Lines 202-204: the authors find that the limit of the breather existence region depends on pump power. This was already predicted (through simulations) in Ref 19 (Fig 6 therein).

12) The authors find that the breathing frequency increases with increasing detuning (Fig 3a). This is the *opposite* trend as what was found in Ref 21 (Fig 5 of the Nature Comm version). Yet both the authors of the current manuscript and that of Ref 21 find it matches simulations. As there is an apparent contradiction, the authors should address it.

13) I like the temporal measurements presented in Fig 4 and obtained through an oversampling technique. This is somewhat a novelty in the microresonator context. Of course, time-resolved observations of the dynamics of dissipative Kerr solitons have been presented many times before that using fiber resonators, so in that sense it is not really a novelty. Yet I think the technique developed by the authors deserves to be presented and maybe discussed in more details in a more specialized journal. Irrespective of the technique, I find that the $\pi/2$ -desynced locked breathing of two breathers presented in Fig 4g,h is, for me, really the only original/novel fundamental observation of the paper. But I do not think that that alone deserves publication in Nature Comm, as it really is a very specialized aspect.

Overall, this is a very detailed, systematic study of breather soliton characteristics in

microresonators, of high technical quality. But a let's-measure-everything-very-accurately approach of something that is now reasonably well known, with two previous high impact publications on the same topic, does not really make the cut for me for publication in Nature Comm.

Reviewer #3 (Remarks to the Author):

In this manuscript the authors experimentally analyze the formation of breathing (temporally oscillating) dissipative solitons in two different microresonator platforms, a MgF2 crystalline resonator and a Si3N4 microring. The dependence of the existence and frequency of these breathers on critical device parameters (i.e. pump power and effective detuning) is investigated. Finally, these experimental results are also interpreted using an analytical and numerical analysis.

The paper addresses an interesting and important question related to the formation of frequency combs in optical microresonators, which is how one can generate stable soliton Kerr combs. Previous theoretical works (Refs. 8 -11) have shown that Kerr solitons can undergo oscillatory instabilities, and such oscillations were experimentally first observed in fiber resonators in Ref. 19. The strength of this paper lies in its detailed and convincing analysis of such breathing instabilities in optical microresonators. As the authors mention, similar observations have been reported very recently in Ref. 20 (Phys. Rev. Lett.) and Ref. 21 (Nature Comm.). The paper is well written, and the reported findings are novel, important, and complementary to previous works. Therefore, I believe this work would be a valuable addition to the literature and I recommend publication after addressing my comments below.

1. In the abstract and main text, the authors mention various times that the breathing instability lacks a comprehensive understanding and that their work provides insight into this process. Although I agree that the measurements convincingly show the different types of oscillatory behavior reported in the theoretical literature, it is not yet clear to me what the physical mechanism is that triggers these oscillations. One key observation is that the breathing emerges when the C-resonance is tuned close to the S-resonance, and their separation is on the order of the resonator linewidth. Could the authors comment more on what they think then physically causes the soliton to destabilize?
2. The first sentence of the introduction (line 24) reads "dissipative solitons are localized solutions of the damped driven nonlinear Schrödinger equation". I find this a strange and narrow definition of dissipative solitons, which are in my opinion localized solutions in any dissipative system. I suggest to rephrase this.
3. On lines 46-50, the authors shortly refer to the recent experimental observations of breathers in microresonators (Refs. 20,21). Although I believe the work of the authors is excellent and merits publication as well, it would be useful to discuss more clearly what the differences are with these other recent works, Refs. 20,21. The current description is rather vague, mentioning that there remain a "variety of open questions", but not discussing them. The authors do say that there is no clear relation between the oscillation frequency and the system parameters, but this does seem to be discussed in Ref. 21.
4. On lines 51-54, the authors say that breather existence allows the prevention of extreme events, referring to Ref. 28. I do not entirely understand what they mean or why they refer to extreme events? As far as I can see, extreme events occur as soon as you come close to a critical threshold, being the moment the stable CW background solution disappears, leading to spatio-temporal chaos. This is shown in more detail in Refs. 8 - 10. So to avoid extreme events, one just steers clear of this chaotic region by staying in the well-characterized region of stable or oscillating solitons. Please clarify or rephrase the statement. I believe the authors want to say that understanding all types of instabilities (not only extreme events) is useful to avoid them in

experiments?

5. In Fig. 2f,g and Fig. 3, a stability map is created in function of the detuning and pump power, both experimentally and numerically. The experiments beautifully show period doubling and period tripling. This is suggestive of more complicated oscillatory dynamics nearby, such as temporal chaos. The authors refer to the existence of such chaos several times, e.g. in the Discussion section. However, the authors never actually show the existence of such temporal chaos. Did the authors explicitly measure such more chaotic dynamics when decreasing the detuning? If so, it would be convincing to show this.

An additional way of backing up the claim of the existence of chaos, would be to compare the generated stability map to Refs. 8,9,19, where analytical expressions are also given to indicate the transition to spatio-temporal chaos. I assume that this will correspond to the boundary of the blue region (MI) in Fig. 3b. However, there also exists the characteristic white region (CW) in Fig. 3b. In this region the attractor corresponding to temporal chaos is destroyed and the system shows transient chaos eventually leading to the CW solution. The experimental data points currently do not show this behavior though, as they stop when breather solutions presumably cease to exist? Can the authors comment on what happens when decreasing the detuning further? Does the system immediately show spatio-temporal chaos or does it have an intermediate region of stable CW operation (which would point to temporal chaos)?

6. Related to the previous comment on chaos, I find the notation "MI" for the region of spatio-temporal chaos rather confusing. In this region of parameter space, the boundary in the stability map does not correspond to a modulational instability. Therefore, I would rather refer to the type of solution in the blue region (\sim spatio-temporal chaos), which is more in line with the notation of the other regions. However, I understand that MI is often used in this context.

7. On lines 181-185, the authors talk about critical slowing down and tipping points. Critical slowing down indeed occurs close to critical transitions. However, the authors don't mention which transition they are thinking about. As they are referring to the change in frequency and period doublings when decreasing the detuning, I assume the authors refer to the global bifurcation in which the chaotic attractor is destroyed? I suggest to explain their reasoning a bit more, certainly as theory has shown many of the transitions/bifurcations that occur.

8. Related to the previous point, the authors refer to Ref. 38, where one analyzes how transitions change when dynamically changing one of the parameters. This is in fact what the authors do here, using the forward and backward tuning method. Could the authors comment on whether the speed of tuning influences their observations of the breathing in parameter space?

9. In fig. 4g the interaction of two breathers is experimentally shown. This measurement nicely correlates with the predicted behavior in Ref. 45. In that work, it was also shown that for larger separation distances, not only the solution with a phase relation of $\pi/2$ exists, but also the in-phase solution is often found. Did the authors find this in-phase solution as well? Moreover, the current experiment shows that the $\pi/2$ solution destabilizes into a single breather. Is this generally the case or do the authors also find stable $\pi/2$ or in-phase breather couples?

10. Some small typos: in several places, articles (the/a) are missing. A few examples: lines 66 (...of optical...), 71-72 (...as Lugiato ...), line 103 (... where breathing ...).

This paper reports a study of the dynamics of breathing (oscillating) solitons in two different platforms. The study is extremely accurate and it provides a complete picture of the dependence of the breathers' stability and frequency in the relevant parameter space made of cavity detuning and pump power. The agreement with numerical simulations based on the LLE is very good. Remarkably, the paper also includes a real time analysis of breathers' dynamics.

Although similar studies appeared recently, one some months ago (Ref. 20) and the other just some weeks ago (Ref. 21), I think that the quality of the research presented here is not diminished by those previous publications, and the paper deserves publication on Nature Communications. However, the publication in particular of [21], which occurred after the submission of the present paper, imposes some changes.

- First, the claim of priority in lines 273-274 must be removed.
 - We have modified this claim in the new version: “we present the first *direct* measurement of the breathing frequency dependence on the laser detuning.” Indeed, in the published version of ref. [21], the detuning is not determined directly and expressed in arbitrary unit and only a trend can be deduced. In this work, all parameters are known allowing the first direct comparison to numerical simulations. We thus believe that this new claim can be maintained.
- Second, a critical comparison with the results of [21] is needed. I found an apparent contradiction between Fig. 3a of the present manuscript and Fig. 5 of [21]. Both figures show a linear dependence of the breathing frequency on the effective cavity detuning, **but the slopes have opposite signs**. Yet, I believe that the contradiction is only apparent, because if one looks at the supplementary material of [21] one realizes that Fig. 5 is just a piece of a larger picture, where for large detuning the breathing frequency increases with the detuning as in Fig. 3a. Since the detuning is measured in units of the cavity linewidth, which is very small in the microresonator considered in Fig. 3a, I guess that the detuning is really very large in that figure. Anyway, a comment on this point would be welcome. On the other hand, the two figures agree in that they show a shift of the breathers toward larger detuning for increasing power.
 - We thank the referee for this comment, and we are also aware of this apparent contradiction. The referee is correct when pointing out that the region simulated in [21] is a narrow region of parameter compared to our range of operation. We reproduced the simulations carried in this paper, which are presented in an additional figure of the method section. We obtained a similar behavior: first the breathing frequency decreases with increased detuning over a narrow range, before the trend reverses to the one we observe. Such a behavior occurs when the driving strength is “below” the CW region (Fig. 4) and the domain of appearance is very small (essentially 1 cavity linewidth). In our experiments, since we are operating above this threshold, the soliton decays before entering this regime. Plus, the system is very thermally unstable in the breathing region, such that we doubt such a small range can be accessed and recorded in our experiment. Also, as pointed out in ref [21], it is surprising that the dominant trend that we report in the present manuscript is not observed. One possible explanation for this fact is that the authors could have observed the appearance of inter-mode breathers, which appear as a result of the avoided modal crossings, and whose oscillation can infringe the fundamental breathing dynamics of pure LLE, or even lead to the DKS decay (Guo et al. ArXiv1705.05003). Another difference from our work is that the authors are working in the multiple-soliton regime, which behavior can be complicated by soliton-soliton interactions (Jang et al., New Journal of Physics, 2015) These comments have been added in the main manuscript and in the method section.

Apart from this I have very few remarks, because the paper is really well written and the results are presented in a very clear way. Just few points:

- - the reference to Fermi-Pasta-Ulam recurrence in the abstract is in my opinion not appropriate, because that feature is not investigated in the paper. The better place for mentioning FPU is the introduction

- We agree with the referee on this point. The mention has been removed in the abstract and the first sentence of the abstract modified.
- - the comment about critical slowing down at lines 181-185 must be either substantiated or removed, because actually I do not see any "criticality" in the observed dynamics
 - The critical transition in the present case corresponds to the decay of the soliton into the homogeneous solution (or to transition to spatiotemporal chaos as shown in ref [36]). This point has been clarified in the manuscript.
- - in "Methods" it would be useful to write the modal equations which are effectively used in the simulations
 - The simulations were performed using the LLE (eq. 1 of the paper). All higher order terms (third order dispersion, modal crossings, Raman effect) were neglected.
- The quality of the figures is very good, but sometimes it is difficult to follow the order of the various subfigures, because they are so many and they are not commented in the text in alphabetic order. For instance, Fig. 1b is commented after Figs. 1i. Moreover, the global caption of Fig. 1 is "Experimental observation of ..." but Fig. 1a is **a simulation**.
 - We would like to thank the referee for this comment. We agree that the panels arrangement was somewhat counterintuitive. We have separated the old Fig. 1 in two figures. The new Figure 1 contains only the simulation panel completed with additional simulation results. The current Figure 2 now shows experimental results, and an experimental scheme has been added to ease the comprehension.

Breathing dissipative solitons in optical microresonators, Lucas, Karpov, Guo, Gorodetski, Kippenberg

The manuscript presents detailed observations of breathing solitons in microresonators. In particular, the breathing frequency and the region of existence of breathers is measured accurately as a function of both detuning and pump power. The dynamics is also resolved in the time domain.

- The technical level of the paper is quite high and the results are clear. I would question however the suitability of the manuscript for Nature Communications. Two previous groups have now published studies of breathers in microresonators, respectively in Phys. Rev. Lett. (Ref 20) and in Nature Communications (Ref 21). Plus breathers have also been observed before in equivalent fiber resonators, as mentioned in the manuscript (Ref 19). I could readily agree that the current manuscript presents a more systematic, maybe more detailed study. Nature Communications is however targeted at "important advances of significance". In light of the other previous studies, this manuscript looks incremental. Observing breathers in microresonators, or even in a Kerr resonator, is not an important advance anymore. The novelty is thus weak. The authors vaguely mention that "a variety of open questions ... and dynamics remain unexplored" (or in the abstract, "lack a comprehensive understanding") but they do not really argue about what is different, novel, or significant in their work in comparison to the two other reported observations of breathers. For all these reasons, I think this paper is more suited for a more specialized journal. Also, I do not think this paper brings more *understanding* (using the working of the abstract) to the breathers. Sure, their parameters are measured accurately and systematically, but no new insights (understanding) is actually brought forward in terms of why they occur, why does the breathing frequency changes the way it does, ...

- We do not agree with the referee. We do think our paper provides useful insights and relations that, first, experimentally validate existing predictions on solitons instabilities, and, bring novel perspectives on the breather properties. For example, we relate the S-resonance to the Hopf bifurcation and the breathing frequency. We believe these observations can prove valuable to the community and help advance in the understanding of breathers. We have rephrased the mentioned sentence in the abstract to "Our observations are in agreement with numerical predictions on dissipative Kerr soliton instabilities ...", have modified the introduction, and stressed this new relation of the S-resonance in the conclusion. We have also emphasized the differences of the present work with the existing literature, namely: First, the direct access and control of the parameters, allowing the mapping of the parameter space. Second, direct time-resolved measurements revealing chaotic breathing, higher periodicity and multi-breather synchronization effects. We believe that an experimental report on these effects of interactions and synchronization still represents a significant advance in Kerr resonators and in the study of nonlinear dynamics in general. Finally, we would like to stress that this study was carried concurrently to [21] – now published in Nature Communications. However, ensuring a more comprehensive analysis and what the referee acknowledges as a "more systematic, maybe more detailed study" takes time.

- Note: Ref 21 is cited as an arXiv. It is now published in Nature Comm 8, 14569 (2017) and the reference should be updated.

- The reference has been updated.

Having said that, there is no doubt that the work is valuable. I therefore provide below a list of more specific points that the authors should address when resubmitting to another journal.

- 1) Both in the abstract and at the beginning of the first paragraph, the authors try to point out the generality of dissipative solitons (in sand dunes, plasmas, ...) yet their definition are very much centered around temporal effect in optics. In the abstract (line 7-8) -> balance of *dispersion* and nonlinearity. Some dissipative solitons (outside of optics) are based on diffraction or even diffusion, not necessarily dispersion. In the 1st paragraph (line 24) -> "dissipative solitons are solutions of the damped NLSE", citing Ref 1. Ref 1 actually explores many systems, not all described by the NLSE, or damped NLSE, that exhibit dissipative

solitons. This should be written more carefully.

o The referee is right, solitons do not only occur as solution of the NLSE. We have made a more general statement in the introduction to qualify soliton as localized structures. Also, we have modified the abstract and introduction significantly to make these points clearer.

- Also, the first sentence of the abstract ending with "ranging from sand dunes, plasma physics, to pulse formation ..." (line 9) does not sound grammatically correct to me. Only one item should be mentioned between "ranging from" and "to".

o We have modified the sentence to: '... plasma physics to hydrodynamics'

- 2) Line 28: "optical frequency combs with large repetition rates in the microwave domain": this is slightly misleading. It could be interpreted as "combs in the microwave domain", whereas the combs considered here are in the optical domain. This could be clarified.

o We have modified the sentence to 'optical frequency combs with large repetition rates in the range of 10s to 1000s gigahertz'

- 3) Lines 32-33: "microresonators appeared as a suitable platform to study DKS properties and dynamics". I wonder why? In some respect, observations of these solitons are much simpler/easier to perform in fiber cavities, where the dynamics are identical but occur over longer timescales (easier to resolve), and several papers have proven that point. In fact, in terms of exciting/observing these solitons, everything is harder in a microresonator.

o In microresonators, we actually show that the breathing is still comparatively slow (MHz range) and the ability to get a single soliton means that the slow oscillations on the comb power reflect directly the breathing dynamics without need of very fast detection. The breathing period is thus spanning a large number of roundtrip (> 4000), which possibly makes the higher periodicity easier to notice than in fiber (breathing period is ~11 roundtrips in [19]). Moreover, the much higher finesse of microresonators makes it possible to reach large driving amplitudes (Up to 900 in this work). These points are now highlighted in the introduction. Furthermore, the fact that fiber cavities typically contain lumped elements that perturb the soliton over one roundtrip, may cause deviations from the LLE. Long fiber cavities are also known to be notably acoustically and thermally sensitive, making their operation non-trivial either.

- 4) Line 34, "field patterns" is ambiguous. This could be interpreted as meaning higher-order spatial modes.

o "field patterns" was changed to "stable inhomogeneous solutions"

- 5) Line 35, it is my understanding that "dark pulses" and "platicons" are synonymous, while here these are presented as different structures.

o We agree with the reviewer that platicons simulated in ref.[12] and dark pulses reported in ref.[17] represent essentially the same structures taking in account the periodic nature of the intracavity waveform. We then changed the sentence from "platicons [12], dark pulses [17], ..." to "dark pulses (platicons) [12,17], ..."

- 6) Line 51, "Understanding the breather regime ... [is] a necessity for applications" -> the only thing that appears necessary is to know where it is, to avoid it. The properties of the breathers per se are not important in the context of envisioned applications.

o The referee is right: for applications, the breathers should generally be avoided. This motivated our study of the parameter space of breather existence. We have modified the wording to "important for applications."

- 7) Line 59, the authors claim they "reveal a link between breathing frequency and driving laser parameters". The existence of that dependence (or link) is not a novelty. Other works on breathers have revealed that. Therefore, it should not be presented as such a grand claim.

o To the best of our knowledge, the existence of this link has been only shown in the published version of ref. [21] (please note that this study wasn't present in the arXiv preprint that we cited), where the authors did not access directly the detuning value of the system. Thus, the operating point (detuning, pump power) is not known and a comparison with simulations cannot be trustworthily made. In our work, all parameters

are precisely known allowing to map out directly the actual (linear, close to unity) relation of the breathing frequency to the effective detuning, and compare accurately to the simulations. To further confirm our experimental results, we have also developed a theoretical analysis for the DKS breathers, which suggests such linear dependence. What is also important, as is pointed out by the referee in the question below, the trend we observed for this dependency is exactly opposite to the one reported in ref. [21]. This means that even though the link shown in our work, according to the reviewer, might not be a novelty, there is an apparent contrariety in the experimental validation of this link, which should be resolved. To conclude, we believe that our work, especially with the latest changes inspired by the referees' comments gives a complete, experimentally-validated overview of the breathing dynamics in different regimes. However, to still address the referee concern, the claim has been rephrased to "revealing the *relation* between breathing frequency and the driving laser parameters".

- 8) Top of page 4, the parameter "f" is referred to as "normalized power". It is however proportional to pump *amplitude* (square root of power). Therefore, I am not sure that referring to it as a normalized *power* is appropriate. Maybe normalized pump amplitude?
 - The referee is correct. We have made the corresponding modifications.
- 9) I was somewhat confused by the references to figures 1f,h and 1g,i. I am not sure what is the difference. 1f,h are presented as "low frequency RF spectra". While 1h starts at zero frequency (low frequency?), 1f starts at 0.6 GHz, which is not so low. Also, what is the difference between f and g (or h and i)? One is the RF spectra and the other is the beatnote, but I am not sure about the exact difference (see also lines 123-126). Are these measured differently? This should be explained better.
 - We thank the referee for making us aware about this ambiguity of the text. "Low frequency" in that context means lower than the repetition rate, which can be significantly high (14 GHz for MgF₂ and 100 GHz for SiN). More specifically, in this set of experiments, the intensity noise in the frequency region where the first breathing tones appear for each system (SiN, MgF₂) has been measured by directly feeding the comb output to a PD. For the SiN platform, such frequency can be indeed considered fairly high (0.8 GHz), however with respect to the other characteristic frequency of 100 GHz, it is relatively low. In the second set of experiments, measurement of the beatnote signal were carried out by using high-speed photodiodes, and (only for SiN) further down-mixing it with a high-frequency LO [13]. Both measurement sets were shown as they constitute characteristic signatures of the Kerr combs, and in particular DKS states. In order to make the text clearer, we have remove the mention 'Low frequency' and distinguished between "RF spectrum of the generated light" and "repetition rate beatnote spectrum". Also, we added a scheme of the experimental setup to help understanding.
- 10) Line 120-122: still with regards to Fig 1, the authors talk about peaks revealing the *harmonics* of the breathing frequency, yet no such harmonics are visible for the SiN platform.
 - The main reason why these peaks are not visible is because the breathing frequency is rather high and the required span to show several harmonics with a decent SNR made the measurement too slow. Also, the 3-dB bandwidth of our low-noise photodetector used for these measurements has only 1 GHz bandwidth, which reduced the measured signal at higher harmonics.
- 11) Lines 202-204: the authors find that the limit of the breather existence region depends on pump power. This was already predicted (through simulations) in Ref 19 (Fig 6 therein).
 - The reference was added when mentioning this point.
- 12) The authors find that the breathing frequency increases with increasing detuning (Fig 3a). This is the *opposite* trend as what was found in Ref 21 (Fig 5 of the Nature Comm version). Yet both the authors of the current manuscript and that of Ref 21 find it matches simulations. As there is an apparent contradiction, the authors should address it.
 - We thank the referee for this comment, and we are also aware of this fact, and it illustrates in particular our advances. We would like to point out that in ref. 21, the precise operating point is not known and thus the agreement is only *qualitative*. In contrast, in our study we employ a modulation response method to obtain the quantitative value of the detuning. Besides, in that work, the authors mention that the dominating trend, that we report in the present manuscript, is not observed, which is "under further

investigation". One possible explanation for this fact is that the authors could have observed the appearance of inter-mode breathers, which appear as a result of the avoided modal crossings, and whose oscillation can infringe the fundamental breathing dynamics of pure LLE, or even lead to the DKS decay (Guo et al. ArXiv1705.05003). Another difference with our work is that the authors are working in the multiple-soliton regime, which behavior can be complicated by soliton-soliton interactions [Jang et al., New Journal of Physics, 2015] These comments have been included in the main manuscript and in the method section, where we added a comparison with the simulation of [21] and the parameters we use in this study, to highlight that the regime of reversed slope occurs over a very narrow range of detuning. Summing up, the trend that is shown in Ref. 21 is not in line with the theoretical prediction, and indeed leaves open questions, corroborating that this work is merely a qualitative work reporting a breather regime – leaving however open what type of breathing behavior is actually observed (i.e. intrinsic to the LLE or due to other effects, as outlined above).

- 13) I like the temporal measurements presented in Fig 4 and obtained through an oversampling technique. This is somewhat a novelty in the microresonator context. Of course, time-resolved observations of the dynamics of dissipative Kerr solitons have been presented many times before that using fiber resonators, so in that sense it is not really a novelty. Yet **I think the technique developed by the authors deserves to be presented and maybe discussed in more details** in a more specialized journal. Irrespective of the technique, I find that the $\pi/2$ -desynced locked breathing of two breathers presented in Fig 4g,h is, for me, really the only original/novel fundamental observation of the paper. But I do not think that that alone deserves publication in Nature Comm, as it really is a very specialized aspect.
 - We added a section in the method section of the paper, to report on the different cases of breathing synchronization we observed.
- Overall, this is a very detailed, systematic study of breather soliton characteristics in microresonators, of high technical quality. But a let's-measure-everything-very-accurately approach of something that is now reasonably well known, with two previous high impact publications on the same topic, does not really make the cut for me for publication in Nature Comm.
 - We highly appreciate that the referee acknowledges the quality of our work. We also sincerely believe that he understands that the amount of work presented in our paper, including careful measurements of breathers in two different platforms, theoretical analysis and simulations, could not be completed in the couple of months separating the appearance of our preprint and the originally cited preprint of ref.[21]. Both works that report breathing dynamics (a new topic in the context of microresonators) have been done concurrently, which we would like to emphasize again here. Nonetheless, we prefer to 'measure-everything-very-accurately' and prioritize the completeness of the study and quality of the data over a let's-quickly-measure-and-be-the-first-to-publish approach, as we are convinced that this is the way scientific research should be carried out. Our work does actually demonstrate the agreement with the LLE, it studies for the first time in the microresonators context the detuning dependence of the breathing, and also reports the time domain dynamics of breather solitons. It completes our understanding of the LLE, and also reports interesting novel synchronization behavior. We believe that "reasonably well known" does not mean the topic has been understood and the scientific understanding is complete. It is precisely this what our manuscript remedies by applying methods that enable to carry out a rigorous comparison to theory, which we believe will be valued by the community.

In this manuscript the authors experimentally analyze the formation of breathing (temporally oscillating) dissipative solitons in two different microresonator platforms, a MgF₂ crystalline resonator and a Si₃N₄ microring. The dependence of the existence and frequency of these breathers on critical device parameters (i.e. pump power and effective detuning) is investigated. Finally, these experimental results are also interpreted using an analytical and numerical analysis.

The paper addresses an interesting and important question related to the formation of frequency combs in optical microresonators, which is how one can generate stable soliton Kerr combs. Previous theoretical works (Refs. 8 -11) have shown that Kerr solitons can undergo oscillatory instabilities, and such oscillations were experimentally first observed in fiber resonators in Ref. 19. The strength of this paper lies in its detailed and convincing analysis of such breathing instabilities in optical microresonators. As the authors mention, similar observations have been reported very recently in Ref. 20 (Phys. Rev. Lett.) and Ref. 21 (Nature Comm.). The paper is well written, and the reported findings are novel, important, and complementary to previous works. Therefore, I believe this work would be a valuable addition to the literature and I recommend publication after addressing my comments below.

- 1. In the abstract and main text, the authors mention various times that the breathing instability lacks a comprehensive understanding and that their work provides insight into this process. Although I agree that the measurements convincingly show the different types of oscillatory behavior reported in the theoretical literature, it is not yet clear to me what the physical mechanism is that triggers these oscillations. One key observation is that the breathing emerges when the C-resonance is tuned close to the S-resonance, and their separation is on the order of the resonator linewidth. Could the authors comment more on what they think then physically causes the soliton to destabilize?
 - To the best of our knowledge, the 62 years old Fermi-Pasta-Ulam recurrence paradox still lacks an intuitive understanding. In nonlinear dynamical systems, it is very difficult to provide the direct physical insights behind the mathematical analyses. In the present case, it is known that breathing dissipative solitons occur upon crossing a Hopf bifurcation, where the attractor destabilizes to a limit cycle [8,9,19]. The emergence of breathing can be related to the fact that the background is getting stronger when reducing the detuning and approaching the resonance, which makes it more modulationally unstable. We agree that our paper does not solve these questions, but we believe nonetheless that it provides useful insights and relations that, first, experimentally validate existing predictions on solitons instabilities, and second, bring novel insights on the breather properties. For example, as pointed out by the referee, we relate the S-resonance to the Hopf bifurcations and as a 'mode' of instability of the soliton. We have rephrased the claim in the abstract and introduction, to "Our observations are in agreement with numerical predictions on dissipative Kerr soliton instabilities..." and stress this new relation of the S-resonance in the conclusion.
- 2. The first sentence of the introduction (line 24) reads "dissipative solitons are localized solutions of the damped driven nonlinear Schrödinger equation". I find this a strange and narrow definition of dissipative solitons, which are in my opinion localized solutions in any dissipative system. I suggest to rephrase this.
 - The referee is right, this point was also noted by Referee #2. We have made a more general statement in the introduction to qualify soliton as localized structures.
- 3. On lines 46-50, the authors shortly refer to the recent experimental observations of breathers in microresonators (Refs. 20,21). Although I believe the work of the authors is excellent and merits publication as well, it would be useful to discuss more clearly what the differences are with these other recent works, Refs. 20,21. The current description is rather vague, mentioning that there remain a "variety of open questions", but not discussing them. The authors do say that there is no clear relation between the oscillation frequency and the system parameters, but this does seem to be discussed in Ref. 21.
 - We have emphasized the differences of the present work with the existing literature, namely: First, the direct access and control of the parameters, allowing the mapping of the parameter space. Second, direct

time-resolved measurements revealing chaotic breathing, higher periodicity and multi-breather synchronization effects. Moreover, in the cited preprint of [21], the trend of the breathing frequency was not measured. We have reworked the introduction to mention the existing work more thoroughly and address the remaining problematics more specifically.

- 4. On lines 51-54, the authors say that breather existence allows the prevention of extreme events, referring to Ref. 28. I do not entirely understand what they mean or why they refer to extreme events? As far as I can see, extreme events occur as soon as you come close to a critical threshold, being the moment the stable CW background solution disappears, leading to spatio-temporal chaos. This is shown in more detail in Refs. 8
 - The numerical investigation of apparition of rogue wave and extreme events in microresonators was carried in [28]. It was demonstrated that such events can occur in the breathing region. In that case, two breathing solitons can interact and collide, which leads to a high amplitude wave. In certain applications, such waves can damage the in-line and measurements equipment. We have added this example in the main text to clarify this point: “allows the prevention of soliton instabilities and extreme events such as solitons collision”
- - 10. So to avoid extreme events, one just steers clear of this chaotic region by staying in the well-characterized region of stable of oscillating solitons. Please clarify or rephrase the statement. I believe the authors want to say that understanding all types of instabilities (not only extreme events) is useful to avoid them in experiments?
 - The referee is right, for applications the breathing should generally be avoided. This motivated our study of the parameter space of breather existence. We have modified the wording to “Characterizing the breather regime is not only of fundamental interest, but important for applications.”
- 5. In Fig. 2f,g and Fig. 3, a stability map is created in function of the detuning and pump power, both experimentally and numerically. The experiments beautifully show period doubling and period tripling. This is suggestive of more complicated oscillatory dynamics nearby, such as temporal chaos. The authors refer to the existence of such chaos several times, e.g. in the Discussion section. However, the authors **never actually show the existence of such temporal chaos**. Did the authors explicitly measure such more chaotic dynamics when decreasing the detuning? If so, it would be convincing to show this.
 - We indeed observed a more complex and irregular breathing behavior. For example, transitions to high periodicity oscillations within a very irregular oscillating dynamic that is hard to convey on a limited figure as the recorded traces are rather long. We have included in the SI the demonstration of P5, P7 and P8 and portion of the more ‘chaotic’ trace.
- An additional way of backing up the claim of the existence of chaos, would be to compare the generated stability map to Refs. 8,9,19, where analytical expressions are also given to indicate the transition to spatio-temporal chaos. I assume that this will correspond to the boundary of the blue region (MI) in Fig. 3b. However, there also exists the characteristic white region (CW) in Fig. 3b. In this region the attractor corresponding to temporal chaos is destroyed and the system shows transient chaos eventually leading to the CW solution. *The experimental data points currently do not show this behavior though, as they stop when breather solutions presumably cease to exist?* Can the authors comment on what happens when decreasing the detuning further? Does the system immediately show spatio-temporal chaos or does it have an intermediate region of stable CW operation (which would point to temporal chaos)?
 - As mentioned in the previous point, we have now added a portion of the trace showing temporal chaos, where no clear periodicity can be identified. It occurs just before the soliton decays upon entering the CW region. In this work, we do not observe the transition to spatiotemporal chaos as demonstrated in [36], since we operate at large pump amplitude and tune backward, we encounter the CW region, where the soliton decays, before being able to reach the region of spatiotemporal chaos. After the soliton decay, the system stays in the low CW branch and the cavity up-switches directly to MI when the detuning is reduced outside the bistable region. Furthermore, to the best of our knowledge, no analytical expressions for the boundaries of the Hopf bifurcation leading to soliton breathing or for the other subdivisions in the breathing domain [9,19], were derived in these references. The boundaries were estimated based on numerical simulations of the spatio-temporal stability of the soliton solution.

- 6. Related to the previous comment on chaos, I find the notation "MI" for the region of spatio-temporal chaos rather confusing. In this region of parameter space, the boundary in the stability map does not correspond to a modulational instability. Therefore, I would rather refer to the type of solution in the blue region (~spatio-temporal chaos), which is more in line with the notation of the other regions. However, I understand that MI is often used in this context.
 - In the various analysis of the LLE, the patterns occurring outside of the bistability region are typically qualified as MI. The domain of bistability is known to allow stable soliton formation and as such is typically treated separately. Yet within this region, various numerical and experimental studies revealed that in a specific region of the parameter space, the system develops spatio-temporal chaos similar to the MI. In this paper, we prefer to stay in line with the existing conventions.
- 7. On lines 181-185, the authors talk about critical slowing down and tipping points. Critical slowing down indeed occurs close to critical transitions. However, the authors don't mention which transition they are thinking about. As they are referring to the change in frequency and period doublings when decreasing the detuning, I assume the authors refer to the global bifurcation in which the chaotic attractor is destroyed? I suggest to explain their reasoning a bit more, certainly as theory has shown many of the transitions/bifurcations that occur.
 - The referee is right, in that case, the critical transition in the present case corresponds to the destruction of the attractor through the decay of the breathing soliton into the background (or to transition to spatiotemporal chaos as shown in ref [36]). This point has been clarified in the manuscript.
- 8. Related to the previous point, the authors refer to Ref. 38, where one analyzes how transitions change when dynamically changing one of the parameters. This is in fact what the authors do here, using the forward and backward tuning method. Could the authors comment on whether the speed of tuning influences their observations of the breathing in parameter space?
 - The tuning speed (compared to the photon lifetime) employed in our experiment can be considered adiabatic. In the present case, the system features bistability throughout the soliton regime and a (Hopf) bifurcation when entering the breathing region. Regarding the bistable region, the tuning speed might play a role when tuning across the CW region, as demonstrated in [36]. Concerning the Hopf bifurcation, we can indeed expect a shift of the transition point as highlighted in [38]. This has been observed in our simulations when comparing the hard soliton seeding in fixed driving conditions with the case where the detuning is swept. It showed a shift of the breathing initiation toward smaller detuning. This may account for some of the deviation in the map showed in Fig. 4.b. However, in our experiments, the slow tuning, combined with the amount of laser noise and thermal jitter of the cavity probably mask these effects. Besides, experimentally, it is not possible to sweep the laser faster while keeping an accurate track of the effective detuning.
- 9. In fig. 4g the interaction of two breathers is experimentally shown. This measurement nicely correlates with the predicted behavior in Ref. 45. In that work, it was also shown that for larger separation distances, not only the solution with a phase relation of $\pi/2$ exists, but also the in-phase solution is often found. Did the authors find this in-phase solution as well? Moreover, the current experiment shows that the $\pi/2$ solution destabilizes into a single breather. Is this generally the case or do the authors also find stable $\pi/2$ or in-phase breather couples?
 - We did observe several cases of two solitons breathing stably in phase, out of phase and in quadrature. These various cases have been added to the SI of the manuscript. Yet, a correlation with the soliton separation and relative breathing phase suggested in [45] could not be deduced.
- 10. Some small typos: in several places, articles (the/a) are missing. A few examples: lines 66 (...of optical...), 71-72 (...as Lugiato ...), line 103 (... where breathing ...).
 - We would like to thank the referee for making us aware of these incorrect phrasings. They have been corrected

REVIEWERS' COMMENTS:

Reviewer #1 (Remarks to the Author):

The authors have made relevant changes in the new version of the paper and in my opinion they have answered satisfactorily to my and the other referees' comments. The crucial point is priority, since in the meantime a paper on the same subject has been published in Nature Communications. After all the discussion between the authors and the referees I am even more convinced that this paper deserves publication in Nature Communications as well, because it is not merely incremental.

Reviewer #2 (Remarks to the Author):

I am happy with the changes made to the manuscript by the authors. As all the other Reviewers seem to consider Nature Communications as a good platform for publication, I will not argue otherwise. I appreciate in particular that this work was done concurrently with the other publications on microresonator breathers. And I agree that the authors' approach is better than a let's quickly-measure-and-be-the-first-to-publish approach ...

Reviewer #3 (Remarks to the Author):

In this resubmission, the authors have addressed all comments well. In my opinion, this is an excellent and very interesting piece of work that certainly merits publication in Nature Communications.